# Online Allocation and Learning in the Presence of Strategic Agents

**Steven Yin**
Department of Industrial Engineering and Operations Research
Columbia University
New York, NY 10027
`sy2737@columbia.edu`

**Shipra Agrawal**
Department of Industrial Engineering and Operations Research
Columbia University
New York, NY 10027
`sa3305@columbia.edu`

**Assaf Zeevi**
Graduate School of Business
Columbia University
New York, NY 10027
`assaf@gsb.columbia.edu`

## Abstract

We study the problem of allocating $T$ sequentially arriving items among $n$ homogeneous agents under the constraint that each agent must receive a pre-specified fraction of all items, with the objective of maximizing the agents' total valuation of items allocated to them. The agents' valuations for the item in each round are assumed to be i.i.d. but their distribution is a priori unknown to the central planner. Therefore, the central planner needs to implicitly learn these distributions from the observed values in order to pick a good allocation policy. However, an added challenge here is that the agents are strategic with incentives to misreport their valuations in order to receive better allocations. This sets our work apart both from the online auction design settings which typically assume known valuation distributions and/or involve payments, and from the online learning settings that do not consider strategic agents. To that end, our main contribution is an online learning based allocation mechanism that is approximately Bayesian incentive compatible, and when all agents are truthful, guarantees a sublinear regret for individual agents' utility compared to that under the optimal offline allocation policy.

## 1 Introduction

A classic sequential resource allocation problem is to allocate $T$ sequentially arriving items to $n$ agents, where each agent must receive a predetermined fraction of the items. The goal is to maximize social welfare, i.e., the agents' total valuation of the items allocated to them. This problem is non-trivial even when the agents' valuations are stochastic and i.i.d. with a known distribution, the main difficulty being that the allocations must be performed in real-time; specifically, an item must be allocated to an agent in the current round without knowledge of their future valuations.

36th Conference on Neural Information Processing Systems (NeurIPS 2022).

A more challenging (and quite useful) extension of the problem which has been the focus of recent literature considers the case where the distribution of the agents' valuations is apriori unknown to the planner. In such settings, algorithms based on online learning can be used to adaptively learn the valuation distribution from observed valuations in previous rounds, and improve the allocation policy over time (see [2, 10, 6, 5] for some examples). However, these mechanisms implicitly assume that the agents report their valuations truthfully, so that the mechanism can directly learn from the reported valuations in order to maximize the social welfare.

Many practical resource allocations settings do not conform with the truthful reporting assumption. In particular, selfish and strategic agents may have an incentive to misreport their valuations if that can lead to individual utility gain (possibly at the expense of social welfare). Hence, an allocation policy that does not take such misreporting incentives into account can incur significant loss in social welfare in presence of strategic agents. For example, consider a simple setting with two agents whose true valuations are i.i.d. and uniformly distributed between 0 and 1. That is,

$$X_1, X_2 \overset{i.i.d.}{\sim} \text{Uniform[0,1]}$$

Each agent is pre-determined to receive an equal fraction of all the items. The optimal welfare maximizing allocation policy is to allocate the item to the agent with higher valuation in (almost) every round. This policy results in $T/3$ expected utility ($\mathbb{E}[X_1|X_1 > X_2]/2$) for each agent, and a social welfare of $2T/3$. However, suppose that the first agent chooses to misreport in the following way: the agent reports a high valuation of 1 whenever her true valuation is in $[0.5, 1]$ and a low valuation of 0 whenever her true valuation is in $[0, 0.5]$. Assuming the other agent remains truthful, this will lead to the first agent receiving all the items in her top $1/2$ quantile, and therefore a significantly increased utility of $3T/8$ ($\mathbb{E}[X_1|X_1 > 0.5]/2$) compared to $T/3$ under truthful reporting. The social welfare however, goes down to $5T/8$ in this case. Thus under the optimal policy, each agent has an incentive to misreport her valuations in order to gain individual utility. The incentives to misreport may be further amplified under an online learning based allocation algorithm that learns approximately optimal policies from the valuations observed in the previous rounds. In such settings, the agents can potentially mislead the online learning algorithm to *learn a more favorable policy* over time by repeatedly misreporting their values.

Motivated by these shortcomings, in this paper, we consider the problem of designing an online learning and allocation mechanism in the presence of *strategic agents*. Specifically, we consider the problem of sequentially allocating $T$ items to $n$ strategic agents. The problem proceeds in $T$ rounds. In each round $t = 1, \ldots, T$, the agents' true valuations $X_{i,t}, i = 1, \ldots, n$ for the $t^{th}$ item are generated i.i.d. from a distribution $F$ *a priori unknown* to the central planner. However, the central planner can only observe a value $\tilde{X}_{i,t}$ reported by each agent $i$, which may or may not be the same as her true valuation $X_{i,t}$ for the item. Using the reported valuations from the current and previous rounds, the central planner needs to make an irrevocable decision of who to allocate the current item. The allocations should be made in a way such that each agent at the end receives a fixed fraction $p_i^*$ of the $T$ items, where $p_i^* > 0, \sum_{i=1}^n p_i^* = 1$. The objective of the central planner is to maximize the total utility of the agents, where utility of each agent is defined as the sum total *true* valuations of items received by the agent.

Our main contribution is a mechanism that achieves both: (a) *Bayesian incentive compatibility*, i.e., assuming all the other agents are truthful, with high probability no single agent can gain a significant utility by deviating from the truthful reporting strategy, and; (b) *near-optimal regret guarantees*, namely, the utility of each individual agent under the online mechanism is "close" to that achieved under the optimal offline allocation policy.

**Organization**   After discussing the related literature in some further detail in Section 2, we formally introduce the problem setting and some of the core concepts in Section 3. Section 4 describes our online learning and allocation algorithm and provides formal statements of our main results (Theorem 1 and Theorem 2). Section 5 and Section 6 provide an overview of the proofs of the above theorems. All the missing details of the proofs are provided in the appendix. Finally, in Section 7 we discuss some limitations and future directions.

## 2 Literature review

Our work lies at the intersection of online learning and mechanism design. From an online learning perspective, our setting is closely related to the recent work on constrained online resource allocation under stochastic i.i.d. rewards/costs (e.g., see [10, 2, 1, 6, 5]). However, a crucial assumption in those settings is that the central planner can observe the true rewards/costs of an allocation, which in our setting would mean that the central planner can observe agents' true valuations of the items being allocated. Our work extends these settings to allow for selfish and strategic agents who may have incentives to misreport their valuations. As discussed in the introduction, unless the online allocation mechanism design takes these incentives into account, selfish agents may significantly misreport their valuations to cause significant loss in social welfare.

Incentives and strategic agents have been previously considered in online allocation mechanism design, however, most of that work has focused on auction design where payments are used as a key mechanism for limiting rational agents' incentives to misreport their valuations. For example, Amin, Rostamizadeh, and Syed [3] study a posted-price mechanism in a repeated auction setting where buyers' valuations are context dependent. Golrezaei, Javanmard, and Mirrokni [12] extend this work to multi-buyer setting, using second-price auction with dynamic personalized reserve prices. Kanoria and Nazerzadeh [17] study a similar problem in a non-contextual setting. (There is also a significant literature that studies learning in repeated auction settings from the bidder's perspective. Since our paper focuses on the central planner's point of view, we omit references to that literature. ) All of the above-mentioned works are concerned with maximizing revenue for the seller, and use money/payments as a key instrument for eliciting private information about the bidder's valuations for the items. In this paper, we are concerned with online allocation without money, and the goal is to maximize each agent's utility.

Recently, there has been some work on studying reductions from mechanism design with money to those without money. Gorokh, Banerjee, and Iyer [13] provided a black-box reduction from any one-shot, BIC mechanism with money, to an approximately BIC mechanism without money. However, their reduction relies crucially on knowing the true value distribution of agents and therefore is not applicable to our setting. Procaccia and Tennenholtz [18] consider a specific (one-round) facility allocation problem and explicitly formulate the idea of designing mechanisms without money to achieve approximately optimal performance against mechanisms with money. Subsequently, there is a series of papers that extended the results on mechanism design without money in a single shot setting, when the bidders' value distribution is unknown [14, 16, 8, 9, 8]. These papers either use a very restricted setting with just two agents, or use very specific/simple valuation functions for the agents. Even in these basic settings, they show that the best one can hope for is a constant approximation to what one can achieve with a mechanism that uses money. It is not clear what kind of regret guarantees such reductions imply in a repeated online learning setting. Therefore, we do not consider such reductions from auction mechanisms with money to be a fruitful direction for achieving our goals of both incentive compatibility and low (sublinear) regret for our online allocation problem.

Finally, in a *repeated* allocation settings with *known* valuation distribution, there are more positive results for truthful mechanism design without money. For example, Guo, Conitzer, and Reeves [15] and later Balseiro, Gurkan, and Sun [7] studied the problem of repeatedly allocating items to agents with known value distributions; both use a state-based "promised utility" framework.

To summarize, to the best of our knowledge, this is the first paper to incorporate strategic agents' incentives in the well-studied online allocation problem with stochastic i.i.d. rewards and unknown distributions. Thus, it bridges the gap between the online learning and allocation literature which focuses on non-strategic inputs, and the work on learning in repeated auctions which focuses on allocation mechanisms that utilize money (payments) to achieve incentive compatibility.

## 3 Problem formulation

**The offline problem**

We first state the offline version of the problem which will serve as our benchmark for the online problem. There is a set of $n$ agents, and a distribution $\boldsymbol{F}$ over $\mathcal{X} := [0, \bar{x}]^n$. Each draw $\boldsymbol{X} \sim \boldsymbol{F}$ from this distribution represents the $n$ agents' valuations of one item: $\boldsymbol{X} = [X_1, \ldots, X_n]$. We assume that

the agents' valuations are i.i.d., i.e.
$$\boldsymbol{F} = F \otimes \ldots \otimes F.$$

A matching policy (aka allocation policy) maps, potentially with some exogenous randomness, a value vector $\boldsymbol{X}$ to one of the agents $i \in \{1, \ldots, n\}$. Specifically, given a realized value vector $\boldsymbol{X} \in [0, \bar{x}]^n$, a (possibly randomized) policy $\pi$ maps $\boldsymbol{X}$ to agent $\pi(\boldsymbol{X}) \in \{1, \ldots, n\}$, with the probability of agent $i$ receiving an allocation given by $\mathbb{P}(\pi(\boldsymbol{X}) = i)$. The offline optimization problem is to find a social welfare-maximizing policy $\pi^*$ such that each agent $i$ in expectation receives a predetermined fraction $p_i^*$ of the pool of items, where $p_i^* > 0, \sum_i p_i^* = 1$. The problem of finding optimal policy can therefore be stated as the following

$$\max_{\pi} \quad \mathbb{E}\left[\sum_{i=1}^{n} X_i \mathbb{1}(\pi(\boldsymbol{X}) = i)\right] \tag{1}$$
$$\text{s.t.} \quad \mathbb{P}(\pi(\boldsymbol{X}) = i) = p_i^* \quad \forall i$$

where the expectations are taken both over $\boldsymbol{X} \sim \boldsymbol{F}$ and any randomness in the mapping made by policy $\pi$ given $\boldsymbol{X}$. Solving the offline problem is non-trivial, as it is an infinite dimensional optimization problem as stated in its' current form in (1). But it turns out to be closely related to Semi-Discrete Optimal Transport, and that the dual of (1) can be written as

$$\min_{\lambda \in \mathbb{R}^n} \mathcal{E}(\lambda, \boldsymbol{F}) \coloneqq \sum_{i \in [n]} \int_{\mathbb{L}_i(\lambda)} (X_i + \lambda_i)\, d\boldsymbol{F}(\boldsymbol{X}) - \lambda^\top p^* \tag{2}$$

where $\mathbb{L}_i$ is what is sometimes referred to as the *Laguerre cell*:

$$\mathbb{L}_i(\lambda) = \left\{\boldsymbol{X} : X_i + \lambda_i > X_j + \lambda_j\, \forall j \neq i, \right\}. \tag{3}$$

Let $\lambda^*(\boldsymbol{F})$ denote an optimal solution to (2). When it is clear from the context, we omit the distribution $\boldsymbol{F}$. It is known that an optimal solution to (1) is given by the following deterministic policy defined by the Laguerre cell partition (Proposition 2.1 [4]):

$$\pi^*(\boldsymbol{X}) = i \text{ for all } \boldsymbol{X} \in \mathbb{L}_i(\lambda^*), i = 1, \ldots, n \tag{4}$$

More generally, we will refer to any policy defined by a Laguerre cell partition as a greedy policy.

**Definition 1** (Greedy allocation policy). *Consider any allocation policy that partitions the domain $[0, \bar{x}]^n$ as $\mathbb{L}_i(\lambda)$ (as defined in (4)) for some $\lambda \in \mathbb{R}^n$. We refer to such a policy as the greedy allocation policy with parameter $\lambda$.*

Note that there are efficient algorithms for solving (2) (see[4]) when the distribution $\boldsymbol{F}$ is known. Therefore we will treat $\lambda^*(\cdot)$ as a black-box that can be computed efficiently for any given input distribution $\hat{F}$.

**The online problem: approximate Bayesian incentive compatibility and regret**

We are interested in the case when items are sequentially allocated over $T$ rounds, and that the distribution $\boldsymbol{F}$ is initially unknown. Specifically, in each round $t = 1, \ldots, T$, the agents' true valuations $\boldsymbol{X}_t = (X_{i,t}, i = 1, \ldots, n)$ are generated i.i.d. from the distribution $\boldsymbol{F}$ *a priori unknown* to the central planner. However, the central planner does not observe $\boldsymbol{X}_t$ but only observes the reported valuations $\tilde{\boldsymbol{X}}_t = (\tilde{X}_{i,t}, i = 1, \ldots, n)$ which may or may not be the same as the true valuations.

An online allocation mechanism consists of a sequence of allocation policies $\pi_1, \ldots, \pi_t$ where the policy $\pi_t$ at time $t$ may be adaptively chosen based on the observed information until before time $t$:

$$\mathcal{H}_t = \{\tilde{\boldsymbol{X}}_1, \ldots, \tilde{\boldsymbol{X}}_{t-1}, \pi_1, \ldots, \pi_{t-1}\}. \tag{5}$$

Given allocation policy $\pi_t$ at time $t$, the agent $i$'s utility at time $t$ is given by

$$u_i(\tilde{\boldsymbol{X}}_t, \boldsymbol{X}_t, \pi_t) = X_{i,t} \mathbb{1}[\pi_t(\tilde{\boldsymbol{X}}_t) = i]$$

If an agent $i$ has already reached his target allocation of $p_i^* T$ items, then he cannot be allocated more items. Note that since the allocation policy may be randomized, for any given value vector $\boldsymbol{X}$, $\pi_t(\boldsymbol{X})$ is a random variable. To ensure truthful reporting in presence of strategic agents, we are interested in mechanisms that are (approximately) Bayesian incentive compatible.

**Definition 2** (Approximate-BIC). *For an online allocation mechanism, let $\pi_t, t = 1, \ldots, T$ be the sequence of allocations when all agents report truthfully, i.e., when $\tilde{\boldsymbol{X}}_t = \boldsymbol{X}_t, \forall t$; and let $\tilde{\pi}_t^i, t = 1, \ldots, T$ be the sequence when all agents except $i$ report truthfully, i.e., $X_{j,t} = \tilde{X}_{j,t}, \forall j \neq i$. Then the online allocation mechanism is called $(\alpha, \delta)$-approximate Bayesian Incentive Compatible if, for all $i = 1, \ldots, n$, with probability at least $1 - \delta$,*

$$\sum_{t=1}^{T} u_i(\tilde{\boldsymbol{X}}_t, \boldsymbol{X}_t, \tilde{\pi}_t^i) - \sum_{t=1}^{T} u_i(\boldsymbol{X}_t, \boldsymbol{X}_t, \pi_t) \leq \alpha$$

*Here, the probability is with respect to the randomness in true valuations $\boldsymbol{X}_t \sim \boldsymbol{F}$ and any randomness in the online allocation policy. For the online policy to be approximate-BIC, the statement should hold for all possible misreporting of valuations $\tilde{X}_{i,t} \neq X_{i,t}$.*

Therefore, if $\alpha$ is small, then an individual agent has little incentive to strategize. Note that approximate-BIC also implies that truthful reporting for all agents constitutes an approximate-Nash equilibrium. Assuming that all agents are truthful, we are also interested in bounding each individual agent's regret.

**Definition 3** (Individual regret). *We define an individual agent $i$'s regret under an online allocation mechanism as the difference between agent $i$'s realized utility over $T$ rounds and the expected utility achieved in the offline expected problem. That is,*

$$Regret_i(T) = T\mathbb{E}[u_i(\boldsymbol{X}, \boldsymbol{X}, \lambda^*)] - \sum_{t=1}^{T} u_i(\boldsymbol{X}_t, \boldsymbol{X}_t, \pi_t). \tag{6}$$

*Here $\pi_1, \ldots, \pi_T$ denote the allocation policies used by the online allocation mechanism in round $t = 1, \ldots, T$.*

Note that since social welfare is given by the sum of all agents' utilities, a bound on individual regret implies a bound on the regret in social welfare of the mechanism.

## 4  Algorithm and main results

We present an online allocation mechanism that is approximately-BIC, and further achieves low regret guarantees on individual regret when all agents are truthful. Our algorithm contains two components: a learner, and a detector. Intuitively, the detector makes sure that the mechanism is approximately BIC, and the learner adaptively learns utility-maximizing allocation policies assuming truthful agents.

The *learner* runs in epochs with geometrically increasing lengths. The starting time of each epoch $k$ is given by $L_k = 2^k, k = 0, 1, \ldots$, which is also when the allocation policy is updated. At the end of each epoch (i.e., at time $t = L_k - 1$ for epoch $k$), the learner takes all the previously reported values from all the agents, and uses them to construct an empirical distribution of the agents' valuations. The learner implicitly assumes truthful agents in its computations. Therefore, since the agents' true valuations are i.i.d., it first constructs a single, one-dimensional empirical distribution $\hat{F}_t$, and then uses it to construct the corresponding $n$-dimensional distribution $\hat{\boldsymbol{F}}_t$:

$$\hat{F}_t(x) = \frac{1}{tn} \sum_{s=1}^{t} \sum_{i=1}^{n} \mathbb{1}[\tilde{X}_{i,s} \leq x] \tag{7}$$

$$\hat{\boldsymbol{F}}_t = \hat{F}_t \otimes \ldots \otimes \hat{F}_t$$

The learning algorithm then solves the offline problem (2) using $\hat{\boldsymbol{F}}_t$, and uses the resulting greedy allocation policy characterized by $\lambda^*(\hat{\boldsymbol{F}}_t)$ to allocate the items in the following epoch.

In parallel to the learner, the *detector* constructs and monitors, in each time step $t$, and for each agent $i$, two empirical distributions. One using the reported valuations from agent $i$: $\bar{F}_t$, and one using the reported valuations from all the other agents: $\tilde{F}_t$. The detection algorithm then computes the supremum between the two empirical CDFs, $\sup_x |\bar{F}_t(x) - \tilde{F}_t(x)|$. If this difference is greater than a predetermined threshold $\Delta_t$, then the detector raises a flag that there has been a violation of truthful reporting and the entire allocation game stops. Otherwise, the process continues.

The threshold $\Delta_t$ needs to be chosen such that if everyone is truthful, then with high probability the detection algorithm will not pull the trigger. At the same time, if someone deviates from truthful reporting significantly, then it should detect this with high probability. The typical concentration result used in comparing empirical CDFs is the Dvoretzky-Kiefer-Wolfowitz (DKW) inequality[11]. However, since a strategic agent can adaptively change its' misreporting strategy, we cannot directly apply the DKW inequality, which assumes i.i.d. samples. Instead, we use martingale version of the DKW inequality (Lemma 1), and use that to choose an appropriate threshold $\Delta_t$. The details are given in Algorithm 1 and Algorithm 2.

---

**Algorithm 1:** Epoch Based Online Allocation Algorithm

**Input:** $T, \delta$
**Initialize:** $\lambda = [0, \ldots, 0], k = 0, K = \log_2(T), L_k = 2^k, k = 0, \ldots, K$;

1   **for** $t \leftarrow 1, 2, 3, \ldots, T$ **do**
2      Observe $\boldsymbol{X}_t$. Run Detection Algorithm (Algorithm 2) with sample set $S = \{\boldsymbol{X}_1, \ldots, \boldsymbol{X}_t\}$,
      and threshold $\Delta_t = 64\sqrt{\frac{1}{t} \log(\frac{256et}{\delta})}$
3      **if** *Detection Algorithm* **Return** *Reject* **then**
4         **Terminates**.
5      **if** *one of the agents* $i \in \{1, \ldots, n\}$ *has reached the allocation capacity* $p_i^* T$ **then**
6         Allocate randomly among agents who have not reached capacity
7      **else**
8         Allocate item using greedy allocation policy $\lambda$
9      **if** $t = L_{k+1} - 1$ **then**
10        Compute $\hat{F}_t$ from samples $\{\boldsymbol{X}_1, \ldots, \boldsymbol{X}_t\}$ as in (7).
11        $\lambda \leftarrow \lambda^*(\hat{\boldsymbol{F}}_t)$
12        $k \leftarrow k + 1$

---

**Algorithm 2:** Detection Algorithm

**Input:** Sample set $S = \{\boldsymbol{X}_1, \ldots, \boldsymbol{X}_t\}$, threshold $\Delta_t$.

1   **for** $i \leftarrow 1, \ldots, n$ **do**
2      Compute $\bar{F}_t(x) = \frac{1}{t} \sum_{s=1}^{t} \mathbb{1}[X_i^s \leq x]$ as the empirical CDF of the samples collected from
      agent $i$
3      Compute $\tilde{F}_t(x) = \frac{1}{t(n-1)} \sum_{s=1}^{t} \sum_{j \neq i} \mathbb{1}[X_j^s \leq x]$ be the empirical CDF of all reported
      values from the other agents.
4      **if** $\sup_x |\tilde{F}_t(x) - \bar{F}_t(x)| \geq \frac{\Delta_t}{2}$ **then**
5         **Return** Reject
6   **Return** Accept

---

Our main results are the following guarantees on incentive compatibility and regret of our online allocation algorithm.

**Theorem 1** (Approximate-BIC). *Algorithm 1 is* $(O(\sqrt{nT \log(nT/\delta)}), \delta)$*-approximate BIC.*

Since truthful reporting constitutes an approximate equilibrium, it is reasonable to then assume that agents will act truthfully. We show the following individual regret bound assuming truthfulness.

**Theorem 2** (Individual Regret). *Assuming all agents report their valuations truthfully, then under the online allocation mechanism given by Algorithm 1, with probability $1 - \delta$, every agent $i$'s individual regret can be bounded as:*

$$Regret_i(T) \leq \frac{4\sqrt{2}}{\sqrt{2} - 1}\sqrt{nT \log(\frac{4n \log_2 T + nT}{\delta})}\bar{x}$$
$$= O(\sqrt{nT \log(nT/\delta)})$$

Showing approximate incentive compatibility, and then guaranteeing regret under the assumption of truthfulness, is an approach commonly seen in the online mechanism design literature (e.g. Theorem 4 in [17]). In the next section, we describe the high level proof ideas for the main results above.

## 5 Proof ideas

**Proof ideas for Theorem 1** We establish that the mechanism is approximately BIC by showing that no single agent has incentive to significantly deviate from reporting true valuations if all the other agents are truthful. The proof consists of two parts. In Step 1, we prove that any significant deviation from the truth can be detected and will lead the mechanism to terminate. In Step 2,3, we prove that in order to achieve a significant gain in utility, an agent indeed has to report values that significantly deviate from the truth.

**Step 1** Assuming that there is only one (unidentified) strategic agent while all the other agents are truthful, we first show that if the detector does not trigger a violation by time $t$ then with high probability, the empirical distribution of valuations reported by the strategic agent is no more than $O(1/\sqrt{t})$ away from the true distribution (Lemma 3). The key observation here is that since the agents' valuations are i.i.d., we can compare their reported values to detect if any single agent's distribution is significantly different from everyone else's. A technical challenge in making statistical comparisons here is that the strategic agent can adaptively change their reporting strategy over time based on the realized outcomes. Therefore, we derive a novel martingale version of the DKW inequality to show concentration of the empirical distribution relative to the true underlying distribution.

**Step 2** In a given round, given the history, the mechanism's allocation policy is a fixed greedy allocation policy given by $\lambda$. If the distribution of strategic agent's reported values differs from the true distribution by at most $\Delta$, then the agent's expected utility gain in that round, compared to reporting truthfully, is at most $O(\Delta)$, (see Lemma 4).

**Step 3** If over $t$ rounds, the distribution of the strategic agent's reported values is at most $\Delta$ away from the true distribution, then the learning algorithm will, with high probability find an allocation policy that is at most $O(\sqrt{n}\Delta)$ away (in terms of individual utility) from what it would have learned if all the agents were truthful instead (see Lemma 2).

To understand the significance and distinction between results in Step 2 vs. Step 3, note that a strategic agent has two separate ways to gain utility. The first is to report valuations in a way that the agent immediately wins more/better items under the central planner's *current* allocation policy. However, since the central planner is updating its' allocation policy over time, the strategic agent can also misreport in a way that benefits its' *future utility*, by "tricking" the central planner into learning a policy that favors him later on. Together, Step 2 and Step 3 show that the agent cannot gain significant advantage over being truthful in either manner.

In many existing works on online *auctions* mechanisms design, where the central planner dynamically adjusts the reserve price over time, these two types of strategic behaviors are in conflict: the agent either sacrifices future utility to gain immediate utility; or sacrifices near-term utility for future utility. The results in those settings therefore often rely on this observation to show approximate incentive compatibility. In our case however, since there is no money involved, it is not clear if such a conflict between short and long term utility exists. Nonetheless, we are able to bound the agents' ability to strategize. Step 2 bounds the agent's short term incentive to be strategic, whereas Step 3 bounds the longer term incentive to be strategic. Combining these steps gives us a proof for Theorem 1.

**Proof ideas for Theorem 2** Recall that here we assume all agents' are truthful. The proof involves two main steps.

**Step 1** We show that uniformly for any $t = 1, \ldots, T$, with high probability, the empirical distribution constructed from the first $t$ samples is close (within a distance of $\tilde{O}(1/\sqrt{nt})$) to the true distribution $F$. Here, the factor of $1/\sqrt{n}$ comes from the fact that in each round we observe $n$ independent samples from the value distribution, one from each of the agents. This also implies that if all the agents are reporting truthfully, then, with high probability, the detector will not falsely trigger.

**Step 2** We show that the allocation policy learned under the empirical distribution estimated from the samples is close to the the optimal allocation policy (Lemma 2). Specifically, after $t$ rounds if the empirical CDF is at most $O(1/\sqrt{nt})$ away from the true distribution, then each agents' expected utility in one round under the learned allocation policy is at most $\sqrt{n/t}$ away (both from above and from below) from the optimal. By using an epoch based approach we can then show that each agents' individual regret is with high probability bounded by $O(\sqrt{nT})$ over the entire planning horizon $T$.

# 6 Proof details

We will now outline our proof in more detail. All missing proofs can be found in the Appendix. First we state the following martingale variation of the well-known Dvoretzky-Kiefer-Wolfowitz (DKW) inequality[11]. This is critical when dealing with strategic agents as they can adapt their strategy over time, resulting in non-independent (reported) values.

**Lemma 1** (Martingale Version of DKW Inequality)**.** *Given a sequence of random variables $Y_1, \ldots, Y_T$, let $\mathcal{F}_t = \sigma(Y_1, \ldots, Y_t), t = 1, \ldots, T$ be the filtration representing the information in the first $t$ variables. Let $F_t(y) := \Pr(Y_t \leq y | \mathcal{F}_{t-1})$, and $\bar{F}_T(y) := \frac{1}{T} \sum_{t=1}^{T} \mathbb{1}[Y_t \leq y]$. Then,*

$$\mathbb{P}\left(\sup_y \left|\bar{F}_T(y) - \frac{1}{T}\sum_{t=1}^{T} F_t(y)\right| \geq \alpha\right) \leq \left(\frac{128eT}{\alpha}\right) e^{-T\alpha^2/128}$$

Next we introduce a new notation to to denote the fraction of allocation that $j$ receives under the greedy allocation policy with parameter $\lambda$ and valuation distribution $\boldsymbol{F}$:

$$p_j(\boldsymbol{F}, \lambda) := \mathbb{P}_{\boldsymbol{X} \sim \boldsymbol{F}}(\boldsymbol{X} \in \mathbb{L}_j(\lambda)).$$

We start with proving Theorem 2, as we will use this to prove Theorem 1 later.

## 6.1 Individual Regret Bound (Theorem 2)

In Algorithm 1, the allocation policy is trained on the empirical distribution constructed from samples. We want to show that this difference between empirical and population distribution will not impair the performance of the resulting allocation policy too much.

**Lemma 2.** *Let $\boldsymbol{G} = G^1 \otimes \ldots \otimes G^n$, and $\boldsymbol{F} = F^1 \otimes \ldots \otimes F^n$ be two distributions over $[0, \bar{x}]^n$ where the marginals on each coordinate are independent. Suppose $\sup_x |F^i(x) - G^i(x)| \leq \Delta \, \forall i$. Let $\lambda = \lambda^*(\boldsymbol{G})$, and $\lambda^* = \lambda^*(\boldsymbol{F})$. Then*

$$|\mathbb{E}_{\boldsymbol{X} \sim \boldsymbol{F}}[u_i(\boldsymbol{X}, \boldsymbol{X}, \lambda)] - \mathbb{E}_{\boldsymbol{X} \sim \boldsymbol{F}}[u_i[\boldsymbol{X}, \boldsymbol{X}, \lambda^*]]| \leq n\Delta\bar{x}$$

**Proof of Theorem 2** Now we have the main ingredients for Theorem 2. We use the DKW inequality to show that the empirical distribution constructed in (7) is close to the true distribution w.h.p.. Then we use Lemma 2 to show that the allocation policy selected by the learner based on the empirical distribution is almost optimal in expectation. The details can be found in the Appendix B.

## 6.2 Approximate-Bayesian Incentive Compatibility (Theorem 1)

Theorem 2 says that online utility of each agent cannot be too far below the offline optimum if everyone behaves truthfully. In order to show approximate-BIC, it suffices to show that the strategic agent cannot gain too much more than the offline optimum. To do so, we need to bound both the short term and longer term incentives for the agent to be strategic.

### 6.2.1 Short term incentive

We start with bounding the short term strategic incentive. We first show that if agent reports from an average distribution that is very different from the truthful distribution, then with high probability Algorithm 2 can detect that. Note that given the strategic agent's strategy in a given round, his reported value is drawn from a distribution potentially different from $F$.

**Lemma 3.** *Fix a time step $t$. Let $\Delta = 64\sqrt{\frac{\log(\frac{256et}{\delta})}{t}}$. Let $F_s, s = 1, \ldots, t$ be the strategic agent's reported value distributions in each time step given the history, i.e., $F_s(x) := \mathbb{P}(\tilde{X}_{i,s} \le x | \mathcal{H}_s)$. If the average distribution $\bar{F} = \frac{1}{t}\sum_{s=1}^{t} F_s$ is such that $\sup_x |\bar{F}(x) - F(x)| \ge \Delta$, then Algorithm 1 will terminate at or before time $t$ with probability at least $1 - \delta$.*

Next, we show that if the agent restricts the reported distribution to not deviate more than $\Delta$ from the true distribution (so that the deviation may go undetected by the detection algorithm), then the potential gain in the agent's utility compared to truthful reporting is upper bounded by $\bar{x}\Delta$. This bounds the agent's incentive to be strategic.

**Lemma 4.** *Fix a round $t$ and a single strategic agent $i$, so that the remaining agents are truthful, i.e., $\tilde{X}_{j,t} = X_{j,t}, \forall j \ne i$. Let $F_r(\cdot)$ denote the marginal distribution of values $\tilde{X}_{i,t}$ reported by the strategic agent $i$ at time $t$ conditional on the history, i.e.,*

$$F_r(x) := \mathbb{P}(\tilde{X}_{i,t} \le x | \mathcal{H}_t).$$

*Suppose that $\sup_x |F(x) - F_r(x)| \le \Delta$. Then, at any time $t$, given any greedy allocation policy $\lambda$,*

$$\mathbb{E}[u_i(\tilde{\boldsymbol{X}}_t, \boldsymbol{X}_t, \lambda)|\mathcal{H}_t] - \mathbb{E}[u_i(\boldsymbol{X}_t, \boldsymbol{X}_t, \lambda)] \le \bar{x}\Delta$$

Note that $F_r$ specifies only the marginal distribution of $\tilde{X}_{i,t}|\mathcal{H}_t$ and not the joint distribution of $(\tilde{\boldsymbol{X}}_t, \boldsymbol{X}_t)|\mathcal{H}_t$. Indeed the above lemma claims that the given bound on utility gain holds for all possible joint distributions as long as the marginal $F_r$ of $\tilde{X}_{i,t}|\mathcal{H}_t$ is at most $\Delta$ away from $F$.

Intuitively, Lemma 3 and Lemma 4 together bound the agent's short term incentive to be strategic: if the agent deviates from the truthful distribution too much, then the mechanism will terminate early and the agent will lose out on all the future utility (Lemma 3); and given any greedy allocation strategy set by the central planner, we have that if the agents deviates within the undetectable range of Algorithm 2, then the gain in utility compared to acting truthfully is small (Lemma 4). Next, we bound an agent's incentive to lie in order to make the mechanism learn a suboptimal greedy allocation policy that is more favorable to the agent.

### 6.2.2 Long term incentive

In order to bound the longer term incentive to be strategic, we want to show that despite agent $i$ being strategic, the central planner can still learn an allocation policy that closely approximates the offline optimal allocation policy. This means that the agent's influence over the central planner's allocation policies is limited.

**Lemma 5.** *Fix a round $T' \le T$ and a strategic agent $i$. If agent $i$ is the only one being strategic, and Algorithm 2 has not been triggered by the end of time $T'$, then with probability $1 - \delta$, $\hat{\lambda} := \lambda^*(\hat{\boldsymbol{F}}_{T'})$ satisfies*

$$\mathbb{E}[u_i(\boldsymbol{X}, \boldsymbol{X}, \hat{\lambda})] - \mathbb{E}[u_i(\boldsymbol{X}, \boldsymbol{X}, \lambda^*)] \le n\Delta_{T'}\bar{x}$$

*where $\Delta_{T'} = 81\sqrt{\frac{1}{nT'}\log(\frac{256e(T')}{\delta})}$ and $\lambda^* = \lambda^*(\boldsymbol{F})$.*

In particular, consider $T' = L_k$. Then the lemma above shows that if an agent was not kicked out by the end of epoch $k - 1$, then with high probability the greedy allocation policy in epoch $k$ will be such that the agents' expected utility by being truthful is close to what he would have received in the offline optimal solution (Lemma 5). We can now combine this result with Lemma 3 and Lemma 4 to bound the utility that any single strategic agent can gain over the entire trajectory.

**Lemma 6.** *If agent $i$ is the only one being strategic, then with probability $1 - \delta$, agent $i$'s online utility is upper bounded by*

$$\sum_{t=1}^{T} u_i(\tilde{\boldsymbol{X}}_t, \boldsymbol{X}_t, \tilde{\lambda}_{k_t}) \le T\mathbb{E}\left[u_i(\boldsymbol{X}, \boldsymbol{X}, \lambda^*)\right] + \frac{286\sqrt{2}}{\sqrt{2}-1}\sqrt{nT\log(\frac{256e\log_2 T}{\delta})}\bar{x}$$

*Here $k_t$ denotes the epoch number that time step $t$ lies in, and $\tilde{\lambda}_{k_t}$ denotes the allocation policy used by the central planner in that epoch.*

**Proof of Theorem 1**   We have already proven Theorem 2 which bounds individual regret defined as the difference $T\mathbb{E}[u_i(\boldsymbol{X}, \boldsymbol{X}, \lambda^*)] - \sum_{t=1}^{T} u_i(\boldsymbol{X}_t, \boldsymbol{X}_t, \lambda_{k_t})$, i.e., the difference between the utility of agent $i$ under the offline optimal policy and that under the allocation policy learned by the algorithm *when all the agents are truthful*. The proof of Theorem 1 follows from plugging in the upper bound on $T\mathbb{E}[u_i(\boldsymbol{X}, \boldsymbol{X}, \lambda^*)]$ from this theorem into Lemma 6, to obtain the desired bound on the expression $\sum_{t=1}^{T} u_i(\tilde{\boldsymbol{X}}_t, \boldsymbol{X}_t, \tilde{\lambda}_{k_t}) - \sum_{t=1}^{T} u_i(\boldsymbol{X}_t, \boldsymbol{X}_t, \lambda_{k_t})$, i.e., on the total gain in utility achievable by misreporting under our mechanism. Further details are in Appendix C.

## 7   Limitations and Future Directions

Although our goal is to develop mechanisms that are robust to selfish and strategic agents, real applications often involve bad faith actors that have extrinsic motivation to behave adversarially. As such, deployment of such resource allocation mechanisms to critical applications requires significant additional validation. In future work we would like to explore the limit of relaxing the i.i.d. assumption that we place on the distribution of valuations across agents. This is a natural relaxation because if one agent thinks the item is good then it's likely that other agents would like the item as well. Furthermore it is also conceivable that agents are heterogeneous and so have different value distributions for the items. However this seems to require a completely different strategy for detecting, and disincentivize strategic behaviors, as we can no longer catch the strategic agent through comparing each agent's reported distribution with that of others.

## Acknowledgement

This work was supported in part by an NSF CAREER award [CMMI 1846792] awarded to author S. Agrawal.

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
