# A Concentration Results

**Lemma 7** (DKW Inequality [11]). *Given i.i.d. samples $X_1, \ldots, X_T$ from a distribution $F$ (cdf), let $\hat{F}_T(x) = \frac{1}{T} \sum_{t=1}^{T} \mathbb{1}[X_t \leq x]$. Then,*

$$\mathbb{P}\left(\sup_x \left|\hat{F}_T(x) - F(x)\right| \geq \alpha\right) \leq 2e^{-2T\alpha^2}$$

## A.1 Proof of Lemma 1

**Lemma 1** (Martingale Version of DKW Inequality). *Given a sequence of random variables $Y_1, \ldots, Y_T$, let $\mathcal{F}_t = \sigma(Y_1, \ldots, Y_t), t = 1, \ldots, T$ be the filtration representing the information in the first $t$ variables. Let $F_t(y) := \Pr(Y_t \leq y | \mathcal{F}_{t-1})$, and $\bar{F}_T(y) := \frac{1}{T} \sum_{t=1}^{T} \mathbb{1}[Y_t \leq y]$. Then,*

$$\mathbb{P}\left(\sup_y \left|\bar{F}_T(y) - \frac{1}{T}\sum_{t=1}^{T} F_t(y)\right| \geq \alpha\right) \leq \left(\frac{128eT}{\alpha}\right) e^{-T\alpha^2/128}$$

*Proof.* This follows from sequential uniform convergence, see Lemma 10,11 in [19], and the fact that indicator functions have fat-shattering dimension 1. $\square$

A more convenient way to use Lemma 1 is the following corollary:

**Corollary 1.** *Given a sequence of random variables $Y_1, \ldots, Y_T$, let $\mathcal{F}_t = \sigma(Y_1, \ldots, Y_t), t = 1, \ldots, T$ be the filtration representing the information in the first $t$ variables. Suppose $F_t(y) = \Pr(Y_t \leq y | \mathcal{F}_{t-1})$, and let $\bar{F}_T(y) := \frac{1}{T} \sum_{t=1}^{T} \mathbb{1}[Y_t \leq y]$. If $\alpha \geq 16\sqrt{\frac{\log(\frac{128et}{\delta})}{T}}$, then with probability $1 - \delta$*

$$\sup_x \left|\bar{F}_T(x) - \frac{1}{T}\sum_{t=1}^{T} F_t(x)\right| \leq \alpha$$

*Proof.*

$$\left(\frac{128eT}{\alpha}\right) e^{-T\alpha^2/128} \leq \delta$$

$$\iff \alpha^2 \geq \frac{128\log(\frac{128eT}{\delta})}{T} + \frac{128}{T}\log(\frac{1}{\alpha^2})$$

$$\Longleftarrow \alpha^2 \geq \frac{256\log(\frac{128eT}{\delta})}{T}$$

$$\iff \alpha \geq 16\sqrt{\frac{\log(\frac{128eT}{\delta})}{T}}$$

$\square$

# B Proof of Theorem 2

## B.1 Proof of Lemma 2

We first state two helper claims. The first one states that for any fixed greedy allocation policy $\lambda$, if the two distributions of valuations are similar, then the final allocation sizes for each receiver will also be close.

**Claim 1.** *Fix a greedy allocation policy $\lambda$. Let $\boldsymbol{G} = G^1 \otimes \ldots \otimes G^n$, and $\boldsymbol{F} = F^1 \otimes \ldots \otimes F^n$ be two distributions over $[0, \bar{x}]^n$ where the marginals in each coordinate are independent. Suppose $\sup_x |F^i(x) - G^i(x)| \leq \Delta \, \forall i$. Then*

$$\sum_j (p_j(\boldsymbol{F}, \lambda) - p_j(\boldsymbol{G}, \lambda))^+ = \sum_j (p_j(\boldsymbol{G}, \lambda) - p_j(\boldsymbol{F}, \lambda))^+ \leq n\Delta.$$

Next we show that if the allocation sizes are similar for two different greedy allocation policies, then the corresponding allocation decisions (domain partitions) are also similar.

**Claim 2.** *Let $\lambda', \lambda$ be any two fixed greedy allocation policies, and $\mathbf{F}$ a distribution over $[0, \bar{x}]^n$. For all $j$, let $\Omega'_j = \mathbb{L}_j(\lambda')$, and $\Omega_j = \mathbb{L}_j(\lambda)$. Suppose $\sum_j (p_j(\mathbf{F}, \lambda') - p_j(\mathbf{F}, \lambda))^+ = \sum_j (p_j(\mathbf{F}, \lambda) - p_j(\mathbf{F}, \lambda'))^+ \leq \Delta$. Then*

$$\mathbb{P}(\Omega'_j \setminus \Omega_j) \leq \Delta \quad \forall j$$

Using these two Claims, we can now prove Lemma 2. The proofs for these two helper Claims follow after the proof of Lemma 2.

*Proof of Lemma 2.* Claim 1 shows that

$$\sum_j (p_j(\mathbf{F}, \lambda) - p_j(\mathbf{G}, \lambda))^+ = \sum_j (p_j(\mathbf{G}, \lambda) - p_j(\mathbf{F}, \lambda))^+ \leq n\Delta$$

Note that by definition of $\lambda^*$ (due to the constraint $\Pr(\Omega_j) = p_j^*$ in problem (1)), we have $p_j(\mathbf{G}, \lambda) = p_j^* = p_j(\mathbf{F}, \lambda^*)$ for all $j$. This means that

$$\sum_j (p_j(\mathbf{F}, \lambda) - p_j(\mathbf{F}, \lambda^*))^+ = \sum_j (p_j(\mathbf{F}, \lambda^*) - p_j(\mathbf{F}, \lambda))^+ \leq n\Delta$$

Now we can apply Claim 2 and conclude that

$$\mathbb{P}(\Omega_j^* \setminus \Omega_j) \leq n\Delta \quad \forall j, \quad \text{and} \quad \mathbb{P}(\Omega_j \setminus \Omega_j^*) \leq n\Delta \quad \forall j, \tag{8}$$

where $\Omega_j = \mathbb{L}_j(\lambda)$ and $\Omega_j^* = \mathbb{L}_j(\lambda^*)$. Therefore,

$$\mathbb{E}_{\mathbf{X} \sim \mathbf{F}}[u_i(\mathbf{X}, \mathbf{X}, \lambda)] - \mathbb{E}_{\mathbf{X} \sim \mathbf{F}}[u_i[\mathbf{X}, \mathbf{X}, \lambda^*]]$$

$$= \int_{\mathbf{X} \in \Omega_i} X_i \, d\mathbf{F}(\mathbf{X}) - \int_{\mathbf{X} \in \Omega_i^*} X_i \, d\mathbf{F}(\mathbf{X})$$

$$\leq \int_{\mathbf{X} \in \Omega_i \setminus \Omega_i^*} X_i \, d\mathbf{F}(\mathbf{X})$$

$$\leq n\Delta\bar{x}$$

Using the same steps as above we can also show that

$$\mathbb{E}_{\mathbf{X} \sim \mathbf{F}}[u_i(\mathbf{X}, \mathbf{X}, \lambda^*)] - \mathbb{E}_{\mathbf{X} \sim \mathbf{F}}[u_i[\mathbf{X}, \mathbf{X}, \lambda]] \leq n\Delta\bar{x}.$$

$\square$

### B.1.1 Proof of Claim 1

We first show variant of Claim 1 where the two distributions only differ in one coordinate:

**Claim 3.** *Fix a greedy allocation policy $\lambda$. Let $\mathbf{G} = G^1 \otimes G^2 \ldots \otimes G^n$, and $\mathbf{F} = F^1 \otimes F^2 \ldots \otimes F^n$ be two distributions over $[0, \bar{x}]^n$ where the marginals in each coordinate are independent. Assume that $\mathbf{G}$ and $\mathbf{F}$ differ only in one coordinate, w.l.o.g. say coordinate $i$. Then, if $\sup_x |F^i(x) - G^i(x)| \leq \Delta$,*

$$\sum_j (p_j(\mathbf{F}, \lambda) - p_j(\mathbf{G}, \lambda))^+ = \sum_j (p_j(\mathbf{G}, \lambda) - p_j(\mathbf{F}, \lambda))^+ \leq \Delta.$$

*Proof.* We start with the distribution $\mathbf{G} = F^1 \cdots F^{i-1} \otimes G^i \otimes F^{i+1} \cdots F^n$ and replace $G^i$ with a distribution $F^i$ to construct $\mathbf{F} = F^1 \otimes F^2 \cdots \otimes F^n$. We will construct $F^i$ in such a way that it is at most $\Delta$ away from $G^i$ and the changes in the allocation proportions are maximized. Note that since $\sum_i p_i(\mathbf{F}, \lambda) = 1$ and $\sum_i p_i(\mathbf{G}, \lambda) = 1$, we know that

$$LHS(\mathbf{F}) := \sum_j (p_j(\mathbf{F}, \lambda) - p_j(\mathbf{G}, \lambda))^+ = \sum_j (p_j(\mathbf{G}, \lambda) - p_j(\mathbf{F}, \lambda))^+ := RHS(\mathbf{F})$$

is always true for any $\mathbf{F}, \mathbf{G}, \lambda$. This means that we can focus on either maximizing either the LHS or the RHS of the above equation. There are two types of $F^i$ that we can use. One is such that

$p_i(\boldsymbol{F}, \lambda) - p_i(\boldsymbol{G}, \lambda) \geq 0$ and the other is $p_i(\boldsymbol{F}, \lambda) - p_i(\boldsymbol{G}, \lambda) < 0$. We can therefore bound the above quantity under these two scenarios separately:

$$\max_{F^i : p_i(\boldsymbol{F},\lambda) - p_i(\boldsymbol{G},\lambda) \geq 0} RHS(\boldsymbol{F}) \iff \max_{F^i : p_i(\boldsymbol{F},\lambda) - p_i(\boldsymbol{G},\lambda) \geq 0} \sum_{j \neq i} (p_j(\boldsymbol{G}, \lambda) - p_j(\boldsymbol{F}, \lambda))^+ \tag{9}$$

$$\max_{F^i : p_i(\boldsymbol{F},\lambda) - p_i(\boldsymbol{G},\lambda) < 0} LHS(\boldsymbol{F}) \iff \max_{F^i : p_i(\boldsymbol{F},\lambda) - p_i(\boldsymbol{G},\lambda) < 0} \sum_{j \neq i} (p_j(\boldsymbol{F}, \lambda) - p_j(\boldsymbol{G}, \lambda))^+ \tag{10}$$

Therefore for the rest of the proof we can focus on bounding the right hand side of (9) and (10).

**Bounding the RHS of (9)**  Let $\tilde{F}(x) \coloneqq (G^i(x) - \Delta)^+ \forall x < \bar{x}, \tilde{F}(\bar{x}) \coloneqq 1$. We claim that the $F^i$ that maximizes $\sum_{j \neq i}(p_j(\boldsymbol{G}, \lambda) - p_j(\boldsymbol{F}, \lambda))^+$, while being at most $\Delta$ away, is $\tilde{F}$. To see this, consider a different distribution $F'$ on the support $[0, \bar{x}]$ such that $\sup_x |F'(x) - G^i(x)| \leq \Delta$. We know that $F'(x) \geq \tilde{F}(x)$.

Later in Claim 4, we show that for any two distributions $G$ and $F$, we can sample $X \sim F$ using $Y$ sampled from $G$ by performing the following transformation:

$$F^{-1}(G^u(Y))$$

where $G^u$ is the random variable defined for distribution $G$ in (19) and $F^{-1} \coloneqq \inf\{x \in \mathbb{R} : F(x) \geq p\}$ denotes the generalized inverse, sometimes also referred to as the quantile function. This is essentially the inverse CDF method applied to a general distribution (instead of a uniformly sampled variable). In particular, let $G^{iu}$ be the following random function:

$$G^{iu}(y) = \begin{cases} G^i(y) & \text{if } G^i(y) = G^i(y_-) \\ \text{Uniform}[G^i(y_-), G^i(y)] & \text{if } G^i(y) > G^i(y_-) \end{cases}$$

Now, denote by $\tilde{\boldsymbol{F}}, \boldsymbol{F}'$, the joint distribution that we get from $\boldsymbol{G}$ on replacing $G^i$ with $\tilde{F}$ and $F'$, respectively. Then, the winning probabilities for the agents in these two cases are:

$$p_i(\tilde{\boldsymbol{F}}, \lambda) = \int_0^{\bar{x}} \prod_{j \neq i} F^j(\tilde{x} + \lambda_i - \lambda_j) d\tilde{F}(\tilde{x})$$

$$= \int_0^{\bar{x}} \mathbb{E}_{G^{iu}(x)} \left[ \prod_{j \neq i} F^j(\tilde{F}^{-1}(G^{iu}(x)) + \lambda_i - \lambda_j) \right] dG^i(x), \tag{11}$$

$$p_j(\tilde{\boldsymbol{F}}, \lambda) = \int_x \tilde{F}(x + \lambda_j - \lambda_i) \prod_{k \notin \{i,j\}} F^k(x + \lambda_j - \lambda_k) dF^j(x), \quad \forall j \neq i \tag{12}$$

$$p_i(\boldsymbol{F}', \lambda) = \int_x \mathbb{E}_{G^{iu}(x)} \left[ \prod_{j \neq i} F^j(F'^{-1}(G^{iu}(x)) + \lambda_i - \lambda_j) \right] dG^i(x), \tag{13}$$

$$p_j(\boldsymbol{F}', \lambda) = \int_x F'(x + \lambda_j - \lambda_i) \prod_{k \notin \{i,j\}} F^k(x + \lambda_j - \lambda_k) dF^j(x), \quad \forall j \neq i \tag{14}$$

Since $\tilde{F}(x) \leq F'(x) \forall x \in [0, \bar{x}]$ by construction, $\tilde{F}^{-1}(p) \geq F'^{-1}(p) \forall p \in [0, 1]$. It's easy to see that (11) $\geq$ (13), and (12) $\leq$ (14). Using this we have

$$\sum_{j \neq i} (p_j(\boldsymbol{G}, \lambda) - p_j(\tilde{\boldsymbol{F}}, \lambda))^+ \geq \sum_{j \neq i} (p_j(\boldsymbol{G}, \lambda) - p_j(\boldsymbol{F}', \lambda))^+.$$

and substituting $F'$ by $G^i$ and again using (11) $\geq$ (13), and (12) $\leq$ (14), we get

$$p_j(\boldsymbol{G}, \lambda) - p_j(\tilde{\boldsymbol{F}}, \lambda) \geq 0 \quad \forall j \neq i,$$
$$p_i(\boldsymbol{G}, \lambda) - p_i(\tilde{\boldsymbol{F}}, \lambda) \leq 0$$

This shows that $\tilde{F}$ is the maximizer of the RHS of (9) among all distributions that are at most $\Delta$ away from $G^i$. Using this we have

$$\max_{F^i : p_i(\boldsymbol{F}, \lambda) - p_i(\boldsymbol{G}, \lambda) \geq 0} \sum_{j \neq i} (p_j(\boldsymbol{G}, \lambda) - p_j(\boldsymbol{F}, \lambda))^+$$

$$= \sum_{j \neq i} (p_j(\boldsymbol{G}, \lambda) - p_j(\tilde{\boldsymbol{F}}, \lambda))^+$$

$$= p_i(\tilde{\boldsymbol{F}}, \lambda) - p_i(\boldsymbol{G}, \lambda)$$

$$= \int_0^{\bar{x}} \prod_{j \neq i} F^j (x + \lambda_i - \lambda_j) \, d\tilde{F}(x) - \int_0^{\bar{x}} \prod_{j \neq i} F^j (x + \lambda_i - \lambda_j) dG^i(x)$$

$$= \int_{x_\Delta}^{\bar{x}} \prod_{j \neq i} F^j (x + \lambda_i - \lambda_j) \, dG^i(x) + \prod_{j \neq i} F^j (\bar{x} + \lambda_i - \lambda_j) \Delta - \int_0^{\bar{x}} \prod_{j \neq i} F^j (x + \lambda_i - \lambda_j) dG^i(x)$$

$$\leq \Delta$$

**Bounding the RHS of (10)** Let $\hat{F}(x) := \min(G^i(x) + \Delta, 1)$. Then we can use the same steps as above for LHS to show that $\hat{F}(x)$ maximizes $\sum_{j \neq i}(p_j(\hat{\boldsymbol{F}}, \lambda) - p_j(\boldsymbol{G}, \lambda))^+$, and that

$$p_j(\boldsymbol{G}, \lambda) - p_j(\hat{\boldsymbol{F}}, \lambda) \leq 0 \, \forall j \neq i$$

$$p_i(\boldsymbol{G}, \lambda) - p_i(\hat{\boldsymbol{F}}, \lambda) \geq 0$$

This shows that $\hat{F}$ is the maximizer of the RHS of (10).From there, we have

$$\max_{F^i : p_i(\boldsymbol{F}, \lambda) - p_i(\boldsymbol{G}, \lambda) < 0} \sum_{j \neq i} (p_j(\boldsymbol{F}, \lambda) - p_j(\boldsymbol{G}, \lambda))^+$$

$$= \sum_j (p_j(\hat{\boldsymbol{F}}, \lambda) - p_j(\boldsymbol{G}, \lambda))^+$$

$$= p_i(\boldsymbol{G}, \lambda) - p_i(\hat{\boldsymbol{F}}, \lambda)$$

$$= \int_0^{\bar{x}} \prod_{j \neq i} F^j (x + \lambda_i - \lambda_j) dG^i(x) - \int_0^{\bar{x}} \prod_{j \neq i} F^j (x + \lambda_i - \lambda_j) \, d\hat{F}(x)$$

$$= \int_0^{\bar{x}} \prod_{j \neq i} F^j (x + \lambda_i - \lambda_j) dG^i(x) - \Delta \prod_{j \neq i} F^j (\lambda_i - \lambda_j) - \int_0^{x^\Delta} \prod_{j \neq i} F^j (x + \lambda_i - \lambda_j) \, dF(x)$$

$$= \int_0^{\bar{x}} \prod_{j \neq i} F^j (x + \lambda_i - \lambda_j) dG^i(x) - \int_0^{x^\Delta} \prod_{j \neq i} F^j (x + \lambda_i - \lambda_j) \, dF(x)$$

$$\leq \Delta$$

where $x^\Delta = F^{-1}(1 - \Delta)$. $\qquad\square$

Using Claim 3 we can now easily prove the original Claim 1.

*Proof of Claim 1.* First we construct the following sequence of distributions where for any two adjacent distributions they only differ on one coordinate.

$$\boldsymbol{G}_0 = \boldsymbol{G} = G^1 \otimes \ldots \otimes G^n,$$
$$\boldsymbol{G}_1 = F^1 \otimes G^2 \otimes \ldots \otimes G^n,$$
$$\boldsymbol{G}_2 = F^1 \otimes F^2 \otimes G^3 \otimes \ldots \otimes G^n,$$
$$\ldots$$
$$\boldsymbol{G}_n = \boldsymbol{F}.$$

Then we can decompose the difference between $p(\boldsymbol{F}, \lambda)$ and $p^*$ into a sum of differences:

$$||p(\boldsymbol{F}, \lambda) - p(\boldsymbol{G}, \lambda)||_1 = ||p(\boldsymbol{G}_n, \lambda) - p(\boldsymbol{G}_0, \lambda)||_1$$

$$\leq \sum_{i=1}^{n} ||p(\boldsymbol{G}_i, \lambda) - p(\boldsymbol{G}_{i-1}, \lambda)||_1$$

$$\leq 2n\Delta$$

where the last step follows from Claim 3. Since $\sum_j (p_j(\boldsymbol{F}, \lambda) - p_j(\boldsymbol{G}, \lambda))^+ = \sum_j (p_j(\boldsymbol{G}, \lambda) - p_j(\boldsymbol{F}, \lambda))^+ = \frac{1}{2}||p(\boldsymbol{F}, \lambda) - p(\boldsymbol{G}, \lambda)||_1$, we have

$$\sum_j (p_j(\boldsymbol{F}, \lambda) - p_j(\boldsymbol{G}, \lambda))^+ = \sum_j (p_j(\boldsymbol{G}, \lambda) - p_j(\boldsymbol{F}, \lambda))^+ \leq n\Delta$$

□

### B.1.2 Proof of Claim 2

*Proof.* WLOG, assume that $\lambda_1 - \lambda_1' \geq \lambda_2 - \lambda_2' \ldots \geq \lambda_n - \lambda_n'$. Let $Q_{jk} = \Omega_j \cap \Omega_k'$ be the "error flow" of items from $j$ to $k$. It is easy to see that for $k < j$, $\boldsymbol{X}_j + \lambda_j > \boldsymbol{X}_k + \lambda_k \implies \boldsymbol{X}_j + \lambda_j' > \boldsymbol{X}_k + \lambda_k'$. Therefore $Q_{jk} = \emptyset$ for $k < j$. Then it follows that

$$\Omega_j' \setminus \Omega_j \subseteq \bigcup_{i:i<j} Q_{ij} \subseteq \bigcup_{i:i<j} \bigcup_{k:k\geq j} Q_{ik}.$$

The right hand side above is the net outflow from the set $\{i : i < j\}$. However, we know that each individual agents' net in flow is $p_j(\boldsymbol{F}, \lambda') - p_j(\boldsymbol{F}, \lambda)$, so we can bound the RHS by

$$\mathbb{P}\left(\bigcup_{i:i<j} \bigcup_{k:k\geq j} Q_{ik}\right) \leq \sum_{i:i<j} p_j(\boldsymbol{F}, \lambda') - p_j(\boldsymbol{F}, \lambda) \leq \Delta$$

□

## B.2 Proof of Theorem 2

*Proof.* Fix an epoch $k$, let $\Delta = \sqrt{\frac{1}{2n(L_k-1)} \log(\frac{2}{\delta})}$, $\hat{\lambda} = \lambda^*(\hat{\boldsymbol{F}}_{L_k-1})$.

$$(\text{Lemma 7}) \qquad \sup_x |\hat{F}_{L_k-1}(x) - F(x)| \leq \Delta \qquad\qquad\qquad \text{w.p. } 1 - \delta/2$$

$$(\text{Lemma 2}) \implies \mathbb{E}[u_i(\boldsymbol{X}, \boldsymbol{X}, \lambda^*)] - \mathbb{E}[u_i(\boldsymbol{X}, \boldsymbol{X}, \hat{\lambda})] \leq n\Delta\bar{x} \quad \forall i \qquad \text{w.p. } 1 - \delta/2 \tag{15}$$

$$(\text{Chernoff bound}) \implies \mathbb{E}[u_i(\boldsymbol{X}, \boldsymbol{X}, \lambda^*)](L_{k+1} - L_k) - \sum_{t=L_k}^{L_{k+1}-1} u_i(\boldsymbol{X}_t, \boldsymbol{X}_t, \hat{\lambda})$$

$$\leq n\Delta\bar{x}(L_{k+1} - L_k) + \bar{x}\sqrt{\frac{(L_{k+1} - L_k)}{2} \log(\frac{2}{\delta})} \qquad\qquad \text{w.p. } 1 - \delta$$

$$= \sqrt{2^k n \log(\frac{2}{\delta})}\bar{x} + \sqrt{2^{k-1} \log(\frac{2}{\delta})}\bar{x} \qquad\qquad\qquad \text{w.p. } 1 - \delta \tag{16}$$

The above bounds the regret in one epoch if the algorithm does not terminate before the epoch ends. It remains to show that the algorithm with high probability does not terminate too early. This involves showing that with high probability, no agent hits their capacity constraint $p_j^* T$ significantly earlier than $T$, and that the detection algorithm does not falsely trigger.

Continuing from (16), for any time step $T' \leq T$, we have

$$T\mathbb{E}\left[u_i(\boldsymbol{X}, \boldsymbol{X}, \lambda^*)\right] - \sum_{t=1}^{T'} u_i(\boldsymbol{X}_t, \boldsymbol{X}_t, \lambda_{k_t})$$

$$\leq \sum_{k=0}^{\log_2 T'} \left[ (L_{k+1} - L_k)\mathbb{E}[u_i(\boldsymbol{X}, \boldsymbol{X}, \lambda^*)] - \sum_{t=L_k}^{L_{k+1}-1} u_i(\boldsymbol{X}_t, \boldsymbol{X}_t, \lambda_k)] \right]$$

$$+ (T - T')\mathbb{E}\left[u_i(\boldsymbol{X}, \boldsymbol{X}, \lambda^*)\right]$$

$$= \sum_{k=1}^{\log_2 T} \sqrt{n2^k \log(\frac{2}{\delta})}\bar{x} + \sqrt{\frac{2^k}{2}\log(\frac{2}{\delta})}\bar{x} + (T - T')\bar{x} \qquad \text{w.p } 1 - \delta\log_2 T$$

$$\leq 2\sqrt{n\log(\frac{2}{\delta})}\bar{x} \sum_{k=1}^{\log_2 T} \sqrt{2^k} + (T - T')\bar{x} \qquad \text{w.p } 1 - \delta\log_2 T$$

$$\leq \frac{2\sqrt{2}}{\sqrt{2}-1}\sqrt{nT\log(\frac{2}{\delta})}\bar{x} + (T - T')\bar{x} \qquad \text{w.p } 1 - \delta\log_2 T \quad (17)$$

where the second inequality follows from (16) and union bound. Now, since there are at most $\log_2(T)$ epochs for any $T' \leq T$, above holds for all epochs and therefore for all $T'$ with probability $1 - \delta\log_2(T)$. Now we show that with high probability, for all $T' \leq T - \frac{2\sqrt{2}}{\sqrt{2}-1}\sqrt{nT\log(\frac{2}{\delta})}$ and for any fixed agent $i$, the constraint of total allocation to agent $i$ to be less than $p_i^* T$ will be satisfied. Note that a byproduct of applying Lemma 2 in (15) is that $|p_i(\boldsymbol{F}, \lambda_k) - p_i^*| \leq n\Delta_{L_k-1}$ (See (8)). Fix a time step $\tau$,

$$\sum_{t=1}^{\tau} \mathbb{1}[\arg\max_j \boldsymbol{X}_j + \lambda_{k_t j} = i]$$

$$\text{(Chernoff)} \quad \leq \sum_{k=1}^{\log_2 \tau} (L_{k+1} - L_k)p_i(\boldsymbol{F}, \lambda_k) + \sqrt{\frac{(L_{k+1} - L_k)}{2}\log(\frac{2}{\delta})} \qquad \text{w.p. } 1 - \delta\log_2 \tau$$

$$\leq \sum_{k=1}^{\log_2 \tau} (L_{k+1} - L_k)(p_i^* + n\Delta_{L_k-1}) + \sqrt{\frac{(L_{k+1} - L_k)}{2}\log(\frac{2}{\delta})} \qquad \text{w.p. } 1 - \delta\log_2 \tau$$

$$(L_k = 2^k) \quad \leq p_i^*\tau + \sum_{k=1}^{\log_2 \tau} \left( \sqrt{n2^k \log(\frac{2}{\delta})} + \sqrt{2^{k-1}\log(\frac{2}{\delta})} \right) \qquad \text{w.p. } 1 - \delta\log_2 \tau$$

$$\leq p_i^*\tau + \frac{2\sqrt{2}}{\sqrt{2}-1}\sqrt{n\tau\log(\frac{2}{\delta})} \qquad \text{w.p. } 1 - \delta\log_2 \tau$$

This means that for all $\tau \leq T - \frac{2\sqrt{2}}{\sqrt{2}-1}\sqrt{nT\log(\frac{2}{\delta})}$, with probability $1 - \delta\log_2 T$,

$$\sum_{t=1}^{\tau} \mathbb{1}[\arg\max_j \boldsymbol{X}_j + \lambda_{k_t j} = i] \leq p_i^* T,$$

Combining above with (17), we have that with probability $1 - 2\delta\log_2 T$, for any fixed $i$, if the algorithm terminates at $T'$ due to allocation limit reached for agent $i$, then $T' \geq \frac{2\sqrt{2}}{\sqrt{2}-1}\sqrt{nT\log(\frac{2}{\delta})}$, so that

$$T\mathbb{E}\left[u_i(\boldsymbol{X}, \boldsymbol{X}, \lambda^*)\right] - \sum_{t=1}^{T'} u_i(\boldsymbol{X}, \boldsymbol{X}, \lambda_{k_t}) \leq \frac{2\sqrt{2}}{\sqrt{2}-1}\sqrt{nT\log(\frac{2}{\delta})}\bar{x} + \frac{2\sqrt{2}}{\sqrt{2}-1}\sqrt{nT\log(\frac{2}{\delta})}\bar{x}$$

Finally, we also have to bound the probability that the detection algorithm falsely triggers. For a given time $t$ and for each $i$, let

$$F_t^i(x) = \frac{1}{t} \sum_{t=1}^{t} \mathbb{1}[X_{i,t} \leq x]$$

$$\tilde{F}_t(x) = \frac{1}{t(n-1)} \sum_{t=1}^{t} \sum_{j \neq i} \mathbb{1}[\boldsymbol{X}_{j,t} \leq x]$$

be the empirical CDF for agent $i$ and the rest of the agents. Since all agents are truthful, using Lemma 7 we have that with probability $1 - \delta$,

$$\sup_x |F_t^i(x) - F(x)| \leq \sqrt{\frac{1}{2t} \log(\frac{2}{\delta})}$$

$$\sup_x |\tilde{F}_t(x) - F(x)| \leq \sqrt{\frac{1}{2t(n-1)} \log(\frac{2}{\delta})}$$

This means that $\sup_x |F_t^i(x) - \tilde{F}_t(x)| \leq \sqrt{\frac{1}{t} \log(\frac{2}{\delta})} \leq 32\sqrt{\frac{1}{t} \log(\frac{256et}{\delta})} = \Delta_t/2$, which means that Algorithm 2 is not triggered by agent $i$. Using union bound, we know that with probability $1 - \delta nT$, the algorithm will not end early because of a false trigger (by any agent).

The result follows by replacing $\delta$ with $\frac{\delta}{n(2\log_2 T + T)}$ and take the union bound over all agents.

$\square$

# C  Proof of Theorem 1

## C.1  Proof of Lemma 3

*Proof.* Let $\alpha = \frac{\Delta}{4}$. We first check that the given condition on $\Delta$ satisfies $\left(\frac{128et}{\alpha}\right) e^{-t\alpha^2/128} \leq \frac{\delta}{2}$ and that $2e^{-2t(n-1)\alpha^2} \leq \frac{\delta}{2}$

$$\left(\frac{128et}{\alpha}\right) e^{-t\alpha^2/128} \leq \frac{\delta}{2}$$

$$\Longleftrightarrow \alpha^2 \geq \frac{128 \log(\frac{256et}{\delta})}{t} + \frac{64}{t} \log(\frac{1}{\alpha^2})$$

$$\Longleftarrow \alpha^2 \geq \frac{256 \log(\frac{256et}{\delta})}{t}$$

$$\Longleftrightarrow \Delta \geq 64\sqrt{\frac{\log(\frac{256et}{\delta})}{t}}$$

$$2e^{-2t(n-1)\alpha^2} \leq \frac{\delta}{2}$$

$$\Longleftrightarrow \alpha \geq \sqrt{\frac{1}{2t(n-1)} \log(\frac{4}{\delta})}$$

$$\Longleftarrow \Delta \geq 64\sqrt{\frac{\log(\frac{256et}{\delta})}{t}}$$

Let $\bar{F}_t(x) = \frac{1}{t} \sum_{s=1}^{t} \mathbb{1}[\tilde{X}_{i,s} \leq x]$ be the empirical CDF of the samples collected from agent $i$. Let $\tilde{F}_t(x) = \frac{1}{(n-1)t} \sum_{s=1}^{t} \sum_{j \neq i} \mathbb{1}[\tilde{X}_{j,s} \leq x]$ be the empirical CDF of all reported values from the other

agents. Let $\bar{F}(x) = \frac{1}{t}\sum_{s=1}^{t} F_s(x)$, where $F_s(x) = \mathbb{P}(\tilde{X}_{i,s} \leq x | \mathcal{H}_s)$. Lemma 1 tells us that with probability $1 - \delta/2$,

$$\sup_x |\bar{F}_t(x) - \bar{F}(x)| \leq \frac{\Delta}{4} \tag{18}$$

Since other agents are truthful, their reported values are independent, and we can use the regular DKW inequality to bound the empirical distribution constructed from their values. Using Lemma 7 we can show that with probability $1 - \delta/2$,

$$\sup_x |\tilde{F}_t(x) - F(x)| \leq \frac{\Delta}{4}.$$

Using union bound, we can conclude that if $\sup_x |\bar{F}(x) - F(x)| \geq \Delta$, then with probability $1 - \delta$:

$$\sup_x |\tilde{F}_t(x) - \bar{F}_t(x)| > \frac{\Delta}{2}$$

which means that Algorithm 2 would have returned Reject. $\qquad\square$

## C.2 Proof of Lemma 4

First we state a technical result on monotone mapping between two distributions. Given a cumulative distribution function $F$, we define the following random function:

$$F^u(y) = \begin{cases} F(y) & \text{if } F(y) = F(y_-) \\ \text{Uniform}[F(y_-), F(y)] & \text{if } F(y) > F(y_-) \end{cases} \tag{19}$$

If $F$ is a continuous distribution then $F^u$ is deterministic and is the same as $F$. However if $F$ contains point masses, then at points where $F$ jumps, $F^u$ is uniformly sampled from the interval of that jump. It is easy to see that $F^u$ has the nice property that if $Y \sim F$, then $F^u(Y) \sim \text{Uniform}[0, 1]$.

**Claim 4.** *Let $G$ be any distribution (cdf) over $\mathcal{X} \subseteq \mathbb{R}$, and $F$ over $\mathcal{Y} \subseteq \mathbb{R}$. Then there exists a unique joint distribution $r$ over $\mathcal{X} \times \mathcal{Y}$ with marginals $G, F$ such that the conditional distribution $r(\cdot|Y)$ has the following* monotonicity *property: define $\bar{x}_r(\cdot), \underline{x}_r(\cdot)$ so that $X \in [\underline{x}_r(Y), \bar{x}_r(Y)]$ almost surely, i.e.,*

$$\bar{x}_r(y) = \inf\{x : \mathbb{P}(X > x | Y = y) = 0\}$$
$$\underline{x}_r(y) = \sup\{x : \mathbb{P}(X < x | Y = y) = 0\},$$

*then*

$$\bar{x}_r(y_1) \leq \underline{x}_r(y_2) \quad \forall y_1 < y_2.$$

*In particular, the random variable $X|Y \sim r(\cdot|Y)$ can be sampled as $G^{-1}(F^u(Y))$, where $F^u$ is the random function defined in (19) and $G^{-1} := \inf\{x \in \mathbb{R} : G(x) \geq p\}$ denotes the generalized inverse, sometimes also referred to as the quantile function.*

The proof of this Claim is in Appendix D.1. Using the above result, we derive the following key result that will provide insight into a strategic agent's best response to a greedy allocation strategy. Note that given a particular marginal distribution $G$ for the agent $i$'s reported values and the true value distribution $F$, there are many potential joint distributions between the true and reported valuations. In the following lemma, we show that the "best" joint distribution among these, in terms of agent $i$'s utility maximization, is the one characterized in Claim 4.

**Claim 5.** *Fix a greedy allocation policy $\lambda$. Let $\boldsymbol{X} \in [0, \bar{x}]^n$ be drawn from $F \otimes \ldots \otimes F$. Fix another distribution $G$ over $[0, \bar{x}]$. Given $\boldsymbol{X}$, define $\tilde{\boldsymbol{X}}^*$ as follows: let $\tilde{X}_i^* = G^{-1}(F^u(X_i))$, and $\tilde{X}_j^* = X_j \,\forall j \neq i$. Let $\mathcal{R}$ be the set of all joint distributions over $[0, \bar{x}]^2$ such that the marginals are $F$ and $G$; and for any $r \in \mathcal{R}$, given $\boldsymbol{X}$ define $\tilde{\boldsymbol{X}}^r$ as follows: $\tilde{X}_i^r \sim r(\cdot|X_i)$, and $\tilde{X}_j^r = X_j \,\forall j \neq i$. Then*

$$\mathbb{E}[u_i(\tilde{\boldsymbol{X}}^*, \boldsymbol{X}, \lambda)] \geq \max_{r \in \mathcal{R}} \mathbb{E}[u_i(\tilde{\boldsymbol{X}}^r, \boldsymbol{X}, \lambda)].$$

*Proof.* First we show that for any joint distribution that is not monotone (i.e., does not have the monotonicity property defined in Claim 4), there is a monotone one that obtains at least as much utility. Suppose $r$ is one such joint distribution that is not monotone, i.e., $\exists x_1 < x_2$, s.t. $\bar{x}_r(x_1) > \underline{x}_r(x_2)$ (as defined in Claim 4). First recall that since $X_j \sim F, \forall j$ are independent, the expected utility can be written as the following:

$$\mathbb{E}[u_i(\tilde{\boldsymbol{X}}^r, \boldsymbol{X}, \lambda)] = \int_0^{\bar{x}} \int_{\underline{x}_r(x)}^{\bar{x}_r(x)} x \prod_{j \neq i} F(\tilde{x} + \lambda_i - \lambda_j) dr(\tilde{x}|x) dF(x)$$

Now consider a pair of values $\tilde{x}_1 > \tilde{x}_2$ such that $(\tilde{x}_1, x_1)$ and $(\tilde{x}_2, x_2)$ has a non-zero probability density under distribution $r$. This pair exists because $\bar{x}_r(x_1) > \underline{x}_r(x_2)$. Then using the fact that for $a, b, c, d > 0, a < b, c < d : ac + bd > ad + bc$, we can see that:

$$x_1 \prod F(\tilde{x}_1 + \lambda_i - \lambda_j) + x_2 \prod F(\tilde{x}_2 + \lambda_i - \lambda_j) < x_1 \prod F(\tilde{x}_2 + \lambda_i - \lambda_j) + x_2 \prod F(\tilde{x}_1 + \lambda_i - \lambda_j)$$

This means that if we exchanged the probability mass between the two conditionals of $x_1, x_2$, the utility would be at least as much as before, if not higher. This means that at least one monotone joint distribution belongs in the set of utility maximizing joint distributions. Since Claim 4 showed that the distribution of $(G^{-1}(F^u(X)), X)$ is the unique joint distribution that is monotone, we conclude that $\tilde{\boldsymbol{X}}^*$ as defined in the lemma statement is indeed utility maximizing. $\square$

**Proof of Lemma 4**

*Proof.* Let $G(x) := (F(x) - \Delta)^+ \forall x < \bar{x}$, $G(\bar{x}) := 1$ be the distribution whose CDF is shifted down from $F$ by $\Delta$. Let $\tilde{r}$ be the utility maximizing joint distribution from Claim 5. Let $\hat{r}, \hat{F}$ be a different pair of joint and marginal distribution such that $\sup_x |F(x) - \hat{F}(x)| \leq \Delta$. We know that $\hat{F}(x) \geq G(x)$ for all $x$. Agent $i$'s utilities for using $\hat{r}$ and $\tilde{r}$ respectively, are:

$$\mathbb{E}_{\hat{r}}[u_i(\hat{\boldsymbol{X}}, \boldsymbol{X}, \lambda)] = \int_0^{\bar{x}} x \int_{\underline{x}_{\hat{r}}(x)}^{\bar{x}_{\hat{r}}(x)} \prod_{j \neq i} F(\hat{x} + \lambda_i - \lambda_j) d\hat{r}(\hat{x}|x) dF(x)$$

$$= \int_0^{\bar{x}} x \mathbb{E}_{F^u(x)} \left[ \prod_{j \neq i} F\left( \hat{F}^{-1}(F^u(x)) + \lambda_i - \lambda_j \right) \right] dF(x) \qquad (20)$$

and

$$\mathbb{E}_{\tilde{r}}[u_i(\tilde{\boldsymbol{X}}, \boldsymbol{X}, \lambda)] = \int x \mathbb{E}_{F^u(x)} \left[ \prod_{j \neq i} F\left( G^{-1}(F^u(x)) + \lambda_i - \lambda_j \right) \right] dF(x) \qquad (21)$$

respectively. Since $\hat{F}(x) \geq G(x)$, we know $\hat{F}^{-1}(p) \leq G^{-1}(p)$. Clearly (20) $\leq$ (21). We conclude that given a greedy allocation policy $\lambda$, true valuation $X_{i,t}$ and truthful agents $j \neq i$ (with $\tilde{X}_{j,t} = X_{j,t}$), reporting $\tilde{X}_{i,t} \sim \tilde{r}(\cdot|X_{i,t})$ is a strategy for agent $i$ that maximizes $\mathbb{E}[u_i(\tilde{\boldsymbol{X}}_t, \boldsymbol{X}_t, \lambda)]$ subject to the marginal distribution constraint $\sup_x |F(x) - F_r(x)| \leq \Delta$. That is,

$$\mathbb{E}_r[u_i(\tilde{\boldsymbol{X}}_t, \boldsymbol{X}_t, \lambda)] \leq \mathbb{E}_{\tilde{r}}[u_i(\tilde{\boldsymbol{X}}_t, \boldsymbol{X}_t, \lambda)] \quad \forall r \text{ s.t. } \sup_x |F_r(x) - F(x)| \leq \Delta$$

It remains to bound the difference $\mathbb{E}_{\tilde{r}}[u_i(\tilde{\boldsymbol{X}}, \boldsymbol{X}, \lambda)] - \mathbb{E}[u_i(\boldsymbol{X}, \boldsymbol{X}, \lambda)]$. First note that $G^{-1}(p) = F^{-1}(p + \Delta)$. Then we have that

$$\mathbb{E}_r[u_i(\tilde{\boldsymbol{X}}, \boldsymbol{X}, \lambda)] - \mathbb{E}[u_i(\boldsymbol{X}, \boldsymbol{X}, \lambda)] \qquad (22)$$

$$= \int_0^{\bar{x}} x \left( \mathbb{E}_{F^u(x)} \left[ \prod_{j \neq i} F\left( F^{-1}(F^u(x) + \Delta) + \lambda_i - \lambda_j \right) \right] - \prod_{j \neq i} F(x + \lambda_i - \lambda_j) \right) dF(x)$$

$$\leq \bar{x} \int_0^{\bar{x}} \left( \mathbb{E}_{F^u(x)} \left[ \prod_{j \neq i} F\left( F^{-1}(F^u(x) + \Delta) + \lambda_i - \lambda_j \right) \right] - \prod_{j \neq i} F(x + \lambda_i - \lambda_j) \right) dF(x) \quad (23)$$

where the inequality follows from the fact that $F^{-1}(F^u(x) + \Delta) \geq x$ w.p.1 for all $x$. To bound the remaining expression in the integral, we can use the fact that since the marginal distribution of $\tilde{x}$ under the joint distribution $\tilde{r}(\tilde{x}, x)$ is $G$, we have

$$\int_0^{\bar{x}} \int_0^{\bar{x}} \prod_{j \neq i} F\left(\tilde{x} + \lambda_i - \lambda_j\right) d\tilde{r}(\tilde{x}|x) dF(x)$$

$$= \int_0^{\bar{x}} \prod_{j \neq i} F\left(x + \lambda_i - \lambda_j\right) dG(x)$$

$$= \int_{x_\Delta}^{\bar{x}} \prod_{j \neq i} F\left(x + \lambda_i - \lambda_j\right) dF(x) + \prod_{j \neq i} F\left(\bar{x} + \lambda_i - \lambda_j\right) \Delta$$

$$\leq \int_{x_\Delta}^{\bar{x}} \prod_{j \neq i} F\left(x + \lambda_i - \lambda_j\right) dF(x) + \Delta \tag{24}$$

where $x_\Delta := F^{-1}(\Delta)$. Similarly,

$$\int_0^{\bar{x}} \prod_{j \neq i} F\left(x + \lambda_i - \lambda_j\right) dF(x)$$

$$= \int_0^{x_\Delta} \prod_{j \neq i} F\left(x + \lambda_i - \lambda_j\right) dF(x) + \int_{x_\Delta}^{\bar{x}} \prod_{j \neq i} F\left(x + \lambda_i - \lambda_j\right) dF(x)$$

$$\geq \int_{x_\Delta}^{\bar{x}} \prod_{j \neq i} F\left(x + \lambda_i - \lambda_j\right) dF(x) \tag{25}$$

Plugging (24) and (25) back to (23), we can now bound the expression in (22), and thereby the profit from strategizing, by $\bar{x}\Delta$.

$\square$

### C.3 Proof of Lemma 5

*Proof.* Let $\bar{F}$ be the average distribution that agent $i$ reported from up to round $T'$: $\bar{F} = \frac{1}{T'} \sum_{t=1}^{T'} F_t$, where $F_t$ is the reported value distribution of agent $i$ in time $t$: $F_t(x) := \mathbb{P}(\tilde{X}_{i,t} \leq x | \mathcal{H}_t)$. Since the the detection algorithm has not been triggered, we can conclude using Lemma 3 that with probability $1 - \delta$,

$$\sup_x |\bar{F}(x) - F(x)| < \Delta := 64\sqrt{\frac{\log(\frac{256eT'}{\delta})}{T'}},$$

$$\text{and} \quad \sup_x |\bar{F}_{T'}(x) - \bar{F}(x)| < \frac{\Delta}{4} = 16\sqrt{\frac{\log(\frac{256eT'}{\delta})}{T'}}.$$

The second inequality holds because the proof of Lemma 3 uses the second inequality to show the first (see Equation 18). Combining the above two steps, we have

$$\sup_x |\bar{F}_{T'}(x) - F(x)| < \frac{\Delta}{4} = 80\sqrt{\frac{\log(\frac{256eT'}{\delta})}{T'}} \qquad \text{w.p. } 1 - \delta. \tag{26}$$

This shows that if the detection algorithm has not been triggered, the empirical CDF of strategic agent's reported values are close to the true CDF. Let $\tilde{F}_{T'}(x) = \frac{1}{(n-1)T'} \sum_{t=1}^{T'} \sum_{j \neq i} \mathbb{1}[X_{j,t} \leq x]$ be the emipircal distriution from all agents other than $i$. We know from Lemma 7 that

$$\sup_x |\tilde{F}_{T'}(x) - F(x)| \leq \sqrt{\frac{1}{2(n-1)T'} \log(\frac{2}{\delta})} \qquad \text{w.p. } 1 - \delta. \tag{27}$$

Combining (26) and (27), we can now bound the error in the combined estimation, $\hat{F}_{T'} = \frac{1}{nT'} \sum_{t=1}^{T'} \sum_{j=1}^{n} \mathbb{1}[\boldsymbol{X}_j^t \leq x]$:

$$\sup_x |\hat{F}_{T'}(x) - F(x)|$$

$$= \sup_x |\frac{1}{n}\bar{F}_{T'}(x) + \frac{n-1}{n}\tilde{F}_{T'}(x) - F(x)|$$

$$= \sup_x |\frac{1}{n}\bar{F}_{T'}(x) - \frac{1}{n}F(x) + \frac{n-1}{n}\tilde{F}_{T'}(x) - \frac{n-1}{n}F(x)|$$

$$\leq \sup_x |\frac{1}{n}\bar{F}_{T'}(x) - \frac{1}{n}F(x)| + \sup_x |\frac{n-1}{n}\tilde{F}_{T'}(x) - \frac{n-1}{n}F(x)|$$

$$\leq 80\sqrt{\frac{\log(\frac{256eT'}{\delta})}{nT'}} + \sqrt{\frac{1}{2nT'}\log(\frac{2}{\delta})} \qquad\qquad \text{w.p. } 1-2\delta$$

$$\leq 81\sqrt{\frac{\log(\frac{256eT'}{\delta})}{nT'}} \qquad\qquad\qquad\qquad \text{w.p. } 1-2\delta \qquad (28)$$

Let $\hat{\boldsymbol{F}}_{T'} = \hat{F}_{T'} \otimes \ldots \otimes \hat{F}_{T'}$, and $\lambda = \lambda^*(\hat{\boldsymbol{F}}_{T'})$, and $\Delta_{T'} = 81\sqrt{\frac{\log(\frac{256eT'}{\delta})}{nT'}}$. Applying Lemma 2 to (28) we have

$$\sup_x |\hat{F}_{T'}(x) - F(x)| \leq \Delta_{T'} \quad \text{w.p. } 1-2\delta$$

$$\text{(Lemma 2)} \implies \mathbb{E}[u_i(\boldsymbol{X}, \boldsymbol{X}, \lambda)] - \mathbb{E}[u_i(\boldsymbol{X}, \boldsymbol{X}, \lambda^*)] \leq n\Delta_{T'}\bar{x}$$

$\qquad\qquad\qquad\qquad\qquad\qquad\qquad\qquad\qquad\qquad\qquad\qquad\qquad\qquad\qquad$ $\square$

### C.4  Proof of Lemma 6

*Proof.* Let $F_t, t = 1, \ldots, T$ be the distributions that agent $i$ reports from in each round given the history, i.e. $\tilde{X}_{i,t}|\mathcal{H}_t \sim F_t$. First we try to bound the utility that the strategic agent can get from a single epoch. Fix an epoch $k$. Suppose $T'$ is the time when either detection algorithm is triggered, or the first time some receiver hits his allocation budget of $p_j^* T$. Let $\tau = \min(T', L_{k+1} - 1)$. We now define three distributions:

$$\bar{F}^1 = \frac{1}{L_k - 1} \sum_{t=1}^{L_k - 1} F_t$$

$$\bar{F}^2 = \frac{1}{\tau - 1} \sum_{t=1}^{\tau - 1} F_t$$

$$\bar{F}^3 = \frac{1}{\tau - L_k} \sum_{t=L_k}^{\tau - 1} F_t$$

These are the average distributions that agent $i$ reported from, averaged across three time periods: $[1, L_k), [1, \tau)$ and $[L_k, \tau)$. In particular, $\bar{F}^3$ is the average distribution that the strategic agent reports from in epoch $k$. From Lemma 3 we know that with probability $1 - 2\delta$:

$$\sup_x |\bar{F}^1(x) - F(x)| \leq 64\sqrt{\frac{\log(\frac{256e(L_k - 1)}{\delta})}{n(L_k - 1)}}$$

$$\sup_x |\bar{F}^2(x) - F(x)| \leq 64\sqrt{\frac{\log(\frac{256e(\tau - 1)}{\delta})}{n(\tau - 1)}}$$

which together means that

$$\sup_x |\bar{F}^2(x) - F(x)| = \sup_x |\frac{L_k}{\tau}(\bar{F}^1(x) - F(x)) + \frac{\tau - L_k}{\tau}(\bar{F}^3(x) - F(x))|$$

$$\implies \sup_x |\bar{F}^2(x) - F(x)| \geq \sup_x |\frac{\tau - L_k}{\tau}(\bar{F}^3(x) - F(x))| - \sup_x |\frac{L_k}{\tau}(\bar{F}^1(x) - F(x))|$$

$$\implies \sup_x |\bar{F}^3(x) - F(x)| \leq \bar{\Delta}_k := \min\left(\frac{128\tau}{\tau - L_k}\sqrt{\frac{\log(\frac{256e(\tau-1)}{\delta})}{n(\tau - 1)}}, 1\right)$$

Note that the last step also uses the fact that the difference between two CDFs cannot be bigger than 1. Let $r$ be any joint distribution for agent $i$'s reported and true valuation $(\tilde{x}, x)$ such that the marginal for the reported valuation is equal to $\bar{F}^3$, i.e.,

$$\bar{X}_{i,t} \sim r(\cdot | X_{i,t}), X_{i,t} \sim F \implies F_r(x) := \mathbb{P}(\bar{X}_{i,t} \leq x) = \bar{F}^3$$

Let $\bar{X}$ denote the reported value vector when $i$ is the only strategic agent and uses $r(\cdot | X_i)$ to pick his reported value: $\bar{X}_j = X_j \,\forall j \neq i$, $\bar{X}_i \sim r(\cdot | X_i)$. Let $\Delta_{L_k-1} = 81\sqrt{\frac{\log(\frac{256e(L_k-1)}{\delta})}{n(L_k-1)}}$. Using this, we have

(Lemma 5) $\implies \mathbb{E}[u_i(\boldsymbol{X}, \boldsymbol{X}, \lambda)] - \mathbb{E}[u_i(\boldsymbol{X}, \boldsymbol{X}, \lambda^*)] \leq n\Delta_{L_k-1}\bar{x}$

(Lemma 4) $\implies \mathbb{E}[u_i(\bar{\boldsymbol{X}}, \boldsymbol{X}, \lambda)] - \mathbb{E}[u_i(\boldsymbol{X}, \boldsymbol{X}, \lambda^*)] \leq n\Delta_{L_k-1}\bar{x} + \bar{\Delta}_k\bar{x}$

(Corollary 1) $\implies \sum_{t=L_k}^{\tau-1} u_i(\tilde{\boldsymbol{X}}_t, \boldsymbol{X}_t, \tilde{\lambda}_{k_t}) - (\tau - L_k)\mathbb{E}[u_i(\boldsymbol{X}, \boldsymbol{X}, \lambda^*)]$

$$\leq (n\Delta_{L_k-1} + \bar{\Delta}_k)\bar{x}(\tau - L_k) + 16\sqrt{(\tau - L_k)\log(\frac{128e(\tau - L_k)}{\delta})}\bar{x} \quad \text{w.p. } 1 - \delta$$

$$\leq 81\sqrt{\frac{n(\tau - L_k)^2}{2(L_k - 1)}\log(\frac{256eL_k}{\delta})}\bar{x} + 144\sqrt{2\tau\log(\frac{256e\tau}{\delta})}\bar{x} \quad \text{w.p. } 1 - \delta$$
(29)

The above is a high probability bound on how much an agent can get in one epoch. We can now bound the strategic agent's utility over the full horizon.

$$\sum_{t=1}^{T'} u_i(\tilde{\boldsymbol{X}}_t, \boldsymbol{X}_t, \tilde{\lambda}_{k_t}) - T'\mathbb{E}[u_i(\boldsymbol{X}, \boldsymbol{X}, \lambda^*)]$$

$$\leq \sum_{k=0}^{\log_2 T'-1}\left[\sum_{t=L_k}^{L_{k+1}-1} u_i(\tilde{\boldsymbol{X}}_t, \boldsymbol{X}_t, \lambda) - (L_{k+1} - L_k)\mathbb{E}[u_i(\boldsymbol{X}, \boldsymbol{X}, \lambda^*)]\right]$$

(Using (29)) $\leq \bar{x}(L_1 - 1) + \sum_{k=0}^{\log_2 T'-1}\left(81\sqrt{\frac{n(L_{k+1} - L_k)^2}{2(L_k - 1)}\log(\frac{256e(L_k - 1)}{\delta})}\bar{x}\right.$

$$\left. + 144\sqrt{2L_{k+1}\log(\frac{256eL_{k+1}}{\delta})}\bar{x}\right) \quad \text{w.p } 1 - \delta\log_2 T$$

$(L_k = 2^k) \leq \bar{x} + \sum_{k=0}^{\log_2 T'-1} 285\sqrt{n2^k\log(\frac{256eT'}{\delta})}\bar{x} \quad \text{w.p } 1 - \delta\log_2 T$

$$\leq \left(\frac{285\sqrt{2}}{\sqrt{2} - 1}\sqrt{nT'\log(\frac{256e}{\delta})} + 1\right)\bar{x} \quad \text{w.p } 1 - \delta\log_2 T$$

The result follows by replacing the original $\delta$ with $\frac{\delta}{\log_2 T}$.

$\square$

# D Auxiliary Proofs

## D.1 Proof of Claim 4

**Claim 4.** *Let $G$ be any distribution (cdf) over $\mathcal{X} \subseteq \mathbb{R}$, and $F$ over $\mathcal{Y} \subseteq \mathbb{R}$. Then there exists a unique joint distribution $r$ over $\mathcal{X} \times \mathcal{Y}$ with marginals $G, F$ such that the conditional distribution $r(\cdot|Y)$ has the following* monotonicity *property: define $\bar{x}_r(\cdot), \underline{x}_r(\cdot)$ so that $X \in [\underline{x}_r(Y), \bar{x}_r(Y)]$ almost surely, i.e.,*

$$\bar{x}_r(y) = \inf\{x : \mathbb{P}(X > x|Y = y) = 0\}$$
$$\underline{x}_r(y) = \sup\{x : \mathbb{P}(X < x|Y = y) = 0\},$$

*then*

$$\bar{x}_r(y_1) \leq \underline{x}_r(y_2) \quad \forall y_1 < y_2.$$

*In particular, the random variable $X|Y \sim r(\cdot|Y)$ can be sampled as $G^{-1}(F^u(Y))$, where $F^u$ is the random function defined in (19) and $G^{-1} := \inf\{x \in \mathbb{R} : G(x) \geq p\}$ denotes the generalized inverse, sometimes also referred to as the quantile function.*

*Proof.* We first prove existence by constructing a joint distribution with the desired marginals and monotonicity, then we show uniqueness.

**Existence.** We will construct the joint distribution by defining the conditional distribution of $X$ given $Y = y$ for every $y$. Note that if $F$ is a continuous distribution, then we can easily construct $r(\cdot|Y = y)$ using the inverse-CDF method:

$$X|y = G^{-1}(F(y))$$

where $G^{-1} := \inf\{x \in \mathbb{R} : G(x) \geq p\}$ is the generalized inverse. This works because $F(Y) \sim$ Uniform[0,1]. If $F$ contains point masses, then $F(Y)$ is no longer uniformly distributed, and the inverse-CDF method does not work. To resolve this, we construct a different random variable $F^u(y)$ for each value $y$. For a given sample $y$, If $F(y) \neq F(y_-)$, let $F^u(y) \sim$ Uniform$[F(y_-), F(y)]$. Otherwise, let $F^u(y) = F(y)$. Now we let

$$X|y = G^{-1}(F^u(y))$$

To see that $X$ sampled using this process has the marginal distribution $G$, we just need to show that $F^u(Y)$ is uniformly distributed. For a given $p$, if $\exists y\, s.t.\, F(y) = p$, then $\mathbb{P}(F^u(Y) \leq p) = \mathbb{P}(F(Y) \leq p) = \mathbb{P}(Y \leq y) = p$. Otherwise that means $\exists y\, s.t.\, p_1 := F(y_-) \leq p$ and $p_2 := F(y) > p$.

$$\begin{aligned}
&\mathbb{P}(F^u(Y) \leq p) \\
&= \mathbb{P}(Y < y) + \mathbb{P}(F^u(y) \leq p|Y = y)\mathbb{P}(Y = y) \\
&= p_1 + \frac{p - p_1}{p_2 - p_1}(p_2 - p_1) \\
&= p
\end{aligned}$$

This construction also satisfies monotonicity, since if $y_1 < y_2$, then $F^u(y_1) \leq F(y_1)$ w.p.1. and $F^u(y_2) \geq F(y_1)$ w.p.1.

**Uniqueness** Now we show uniqueness. For a given $(x, y)$ pair, suppose $x < \bar{x}_r(y)$. Then from monotonicity we know $\underline{x}_r(y') \geq \bar{x}_r(y) > x$ for all $y' > y$, which implies that

$$\mathbb{P}_r(X \leq x, Y \leq y) = G(x).$$

If $x \geq \bar{x}_r(y)$, then from monotonicity we know $\bar{x}_r(y') \leq \bar{x}_r(y) \leq x$ for all $y' < y$, which implies that

$$\mathbb{P}_r(X \leq x, Y \leq y) = F(y)$$

Since $G$ and $F$ are fixed, we have shown that all joint distributions $r$ with monotonicity and the required marginals are the same. $\square$