# OpenReview forum: "Online Allocation and Learning in the Presence of Strategic Agents"
_NeurIPS.cc/2022/Conference — NeurIPS 2022 Accept_

### Official Review · Reviewer_x5pZ · 2022-06-27

**Rating:** 7
**Confidence:** 4
**Soundness:** 4 excellent
**Presentation:** 3 good
**Contribution:** 3 good

**Summary:**

This work considers an online allocation problem with learning and strategic agents. Previous works already studied this problem, but proposed BIC policies either using payments/transfers between the players and the central planner, or assuming that the planner knows beforehand the valuation distribution of the players. This work is thus the first to propose an approximate BIC policy, with sublinear regret, when neither of these conditions hold.

--------

After the discussion with the authors and reviewers, I decided to raise my score as I believe this paper to be well written and to deal with an interesting problem. As suggested in the discussion with the authors, I think that adding a rigorous result stating that a no regret BIC policy cannot be derived in the heterogeneous setting without further assumption.

**Questions:**

My main questions to the authors are related to my concerns mentioned above:

1) Could you show some impossibility result on the heterogeneous case (without payment or distribution knowledge), or do you believe that a good BIC policy is possible in that case?
2) Can you show some guarantees on the respect of the predetermined fractions p_i^* ?

I also have some other questions:
3) The proposed policy is robust to the deviation of a single player. It even seems it could be extended to the possible deviation of a significant fraction of players (eg as in the Byzantine problem). It however might lead to considerable computational costs. Could you develop on this point?

----- Minor comments -----

4) After Definition 2, it could be great to relate the notion of Approximate-BIC to approximate-Nash equilibria
5) The algorithm works in epochs of doubling sizes. Why is it needed? It seems the same algorithm could work with constant epoch size 1 (eg anytime)?


**Limitations:**

No concern

**Strengths And Weaknesses:**

I find this work interesting and overall well written. The considered problem is of interest, since the central planner might have very small leverages in real life situations when allocating resources to multiple agents.

The proposed algorithm is rather simple and natural. It mostly relies on the DKW concentration inequality.

As a main weakness, the proposed algorithm crucially relies on the fact that the players are homogeneous, ie have the same valuation distribution. The case of heterogeneous seems much harder as mentioned by the authors in the conclusion. Actually, I believe that some impossibility result can be shown for heterogeneous players when the player neither knows the players' distribution nor can use payments. Showing such a result would significantly support this work (and I believe it is not that hard).
For a similar reason, the main result claims that truthful bidding by the players is an approximate Nash equilibrium, but I would believe that drastically different strategies could also be approximate Nash equilibria (actually almost any strategy where the players report the same fake distribution).

A second concern I have: the current paper is lacking some theoretical guarantee on how the repartition of the items by the central planner respects the predetermined fraction p_i^*. Having some guarantee on the differences between p_i^* and \hat{p}_i would make the analysis of the policy complete in my opinion.

---

> ### Author Response · Authors · 2022-08-02
> **Thank you for your comments**
>
> Thank you for your comments. We will address your questions and comments in order.
>
> Question 1: Our toy example given in the Introduction section is an intuitive counter example to the existence of a BIC policy when agents are heterogeneous. In this example, the two agents’ true distributions are i.i.d. Uniform[0,1]. However agent 1 can instead report 1 whenever his valuation is above 0.5 and 0 when it’s below 0.5. This will increase his expected utility, but there is no way for the central planner to know if the reported distribution (½ mass at 0 and ½ mass at 1) is his true distribution or not.
>
> Question 2: Our online algorithm does indeed satisfy the target distribution constraints exactly (assuming that $p^*_i T$ are integers). Once one of the agents has been allocated $p^*_iT$ items, the algorithm will simply allocate the rest of the items in an arbitrary order to make sure that every agent receives his target allocation. This will not change the guarantees on the regret bound. We have clarified this part of the algorithm in our revision.
>
> Question 3: This suggestion of exploring Byzantine tolerance type guarantees in an online learning setting is super interesting. We are not aware of any existing papers in this direction. However this is beyond the scope of the current paper.
>
> Question 4: Thank you for the suggestion. Approximate BIC indeed implies that truthful reporting is an approximate nash-equilibrium. We have added this to the revised version of the paper.
>
> Question 5: The choice of epochs with doubling sizes is intentional. The intuition is that this reduces the number of times that the central planner updates the allocation policy. This is useful because the more times that the central planner makes an update, the more opportunities that the strategic agent has to “trick” the central planner into picking a bad policy. However, we do not have a negative result to show that size 1 epochs can not work.

---

> > ### Comment · Reviewer_x5pZ · 2022-08-03
> > **Thanks for your answer**
> >
> > I thank the authors for their answer to the different reviews. I agree that this example gives the intuition that no BIC policy can be proposed in the heterogeneous case. I think that a rigorous proof and claim of this fact would be a bonus to the current version of this work (and that adding such a result could be possible using your counterexample).

---

### Official Review · Reviewer_XRjf · 2022-07-11

**Rating:** 5
**Confidence:** 3
**Soundness:** 3 good
**Presentation:** 4 excellent
**Contribution:** 3 good

**Summary:**

The paper studies a repeated online allocation of a single type of good to agents who have the same but unknown valuation distribution for the good. Unlike the prior work, the paper assumes the agents are strategic and no monetary payments can be used to incentivize truthfulness.  The goal of the allocation is to maximize social welfare. The paper provides an algorithm that satisfies approximate Bayesian incentive compatibility (i.e. no single agent can deviate from truthfulness to gain much if the other agents are truthful) and also achieves no regret (the social welfare will approach the highest possible welfare as the number of allocation rounds goes to infinity).

**Questions:**

-- Can the authors propose a motivating example to justify repeated strategic interactions where monetary compensation is not allowed to incentivize truthfulness?

-- Can the authors provide more insight on how the offline optimal allocation is solved? Does this require new tools or the problem has been solved by prior work? Why the allocation is called "Greedy" in Definition 1?

-- The regret over all agents grow as $n\sqrt{nT}$. Is this standard? Are there any lower bounds?

-------------------------------------
Minor comments:
-------------------------------------
-- Please sort the numbers for the references (e.g., line 27) in the paper.

-- Change "Literature Review" to "Literature review" to be consistent with other section headers.

-- Is there any reason to have an upper bound of $\bar{x}$ for valuations instead of 1?

**Limitations:**

The discussion of the limitations is sufficient.

**Strengths And Weaknesses:**

-------------------------------------
Strengths:
-------------------------------------
-- The paper studies a novel allocation setting where strategic interactions are allowed by the agents but the allocator cannot use payments to incentivize truthfulness.

-- The exposition and presentation of the paper were excellent. The setting was clearly discussed. And the high-level ideas of the algorithms/proofs can be easily followed throughout the paper.

-------------------------------------
Weaknesses:
-------------------------------------
-- The setting of the paper is rather strange. Unlike the traditional online learning setting, there is no immediate feedback to the allocator on the quality of the allocation at each round. Hence, I am not sure about the applicability of the results in any interesting domains.

-- The analysis does not seem to provide any interesting insights. The paper is interested in enforcing Bayesian incentive compatibility and the authors compare the reported bids of any specific user to the rest to check whether the user is untruthful (given that the users all have the same valuation distribution). This might be okay if to goal is to detect untruthfulness. But the mechanism simply stops if untruthfulness is detected. The analysis of the regret is only for the case that all agents are truthful. This also significantly weakens the guarantees provided by the paper.

---

> ### Author Response · Authors · 2022-08-02
> **Thank you for your comments**
>
> Thank you for your comments. We will respond to the claimed weaknesses and the questions in order.
>
> Weakness 1 (setting): The reviewer might be thinking of the online learning with bandit feedback setting. In a bandit feedback setting, the central planner only observes the reward for that round after he has picked an action, and the central planner has no idea what the reward would be before the action is chosen. In our paper, the central planner sees all the reported valuations before making the decision and therefore knows exactly how much “reward” (utility) he will get for any possible action he might pick for that round (assuming truthful reporting). That is why we do not have a separate step of revealing the reward in our setting. This is in fact quite common in most online optimization and online allocation literature. One possible application of this would be allocating unsold groceries to food banks, or allocating blood donors to blood banks. See also our response to Question 1 below.
>
> Weakness 2 (analysis): Our result essentially shows that for rational agents, the incentive to cheat is very limited. The reviewer is correct in pointing out that if there is one adversarial (irrational) agent , then he can hurt the utility of the other agents by a lot. A stronger result would guarantee all truthful agents’ utilities despite the presence of 1 or maybe a small number of strategic agents. It is unclear if such a result is possible in our setting, but this is certainly an interesting direction to explore in future work. In fact one of the other reviewers also suggested a similar direction. We do want to point out that showing approximate BIC, and then showing regret bounds under truthfulness, is an approach used in prior works (see for example Kanoria, Nazerzadeh 2021)
>
> Question 1: Allocating food to food banks is one such example. Feeding America is an organization that takes food donations and allocates food to food banks around the country. In this setting, the central planner cannot use real money to elicit private valuations for the food from the food banks. Although technically not required by law to not use real money, there is strong incentive to keep the system this way since the alternative would be against the non-profit nature of the program.
>
> Question 2: As explained in Section 3, the offline allocation problem is a semi-discrete Optimal Transport problem, which has known efficient algorithms. In particular, Equation 2 is a convex optimization problem with n variables. The difficulty of computing Equation 2 is the evaluation of the objective function. However, efficient stochastic optimization methods have been proposed (see Aude et al 2016). The allocation policy based on Laguerre cells is referred to as “greedy” because it has the form of allocating the item to whoever has the highest “score”.
>
> Question 3: Our focus is on bounding individual regret of the agents which is $\sqrt{nT}$. This also implies $n\sqrt{nT}$ overall regret bound. It is not clear if this is a tight bound on the overall regret, although we currently do not have strong reasons to believe either way.
>
> Minor comment 3: Replacing $\bar x$ with $1$ would not affect the analysis or the results of the paper.

---

> > ### Comment · Reviewer_XRjf · 2022-08-07
> > **Thanks for the response**
> >
> > I just want to let you know that I have read the response and appreciate the effort you made to answer my concerns.

---

### Official Review · Reviewer_jFe4 · 2022-07-11

**Rating:** 6
**Confidence:** 4
**Soundness:** 4 excellent
**Presentation:** 3 good
**Contribution:** 3 good

**Summary:**

The paper studies an online allocation setting where arriving items
are allocated among $n$ agents who are "promised" a prespecified
fraction of the items. The goal of the planner is to maximize the
agents' total valuation. As opposed to the standard setting, the
planner not only does not know the realized valuation at each period,
but also the distribution of the values (which are assumed to be iid
across agents and time). Furthermore, the agents are strategic and
misreport their values to affect future allocations. The paper
proposes an online-learning based allocation mechanism that is
approximately BIC and achieves sublinear regret (under truthful
reporting) against the optimal allocation policy that knows the
distribution.


**Questions:**

While the offline problem requires that the allocation satisfy the
promised fraction $p_i^*$ exactly, there is no specific constraint
placed on the online problem. In particular, while the algorithm
ensures that all but one agent's allocation is below the capacity
$p_i^* T$, is it possible to show that the allocations of all agents
are close to $p_i^*T$, given the regret bounds?

**Strengths And Weaknesses:**

The paper studies an important resource allocation problem inspired
from practical considerations, and proposes a mechanism to decrease
the informational requirements on the planner. If the common iid
distribution were known to the planner, the paper shows that the
problem reduces to an offline optimization problem related to an
optimal transport problem, whose solution is parametrized by the dual
variables $\lambda$. Thus, the online problem is reduced to learning
the optimal dual values $\lambda$ from the reports, while ensuring
that the reports remain truthful.

The main insight behind the mechanism is to use the reports of other
agents to detect if an agent is reporting truthfully, and if not, stop
the allocation. This uses the fact that the valuation distributions
are not only independent, but also identical, a major weakness of the
approach in terms of its applicability. This approach also bears
similarity to implementation theory (Caillaud & Roberts 2005, Maskin
1998), and the reader would be helped by connecting to this
literature.

To summarize, the paper proposes an interesting mechanism to overcome
the ignorance of the valuation distribution. However, this approach is
somewhat hampered by its reliance on the iid assumption. In
particular, the approach fails even if the valuation distribution is
independent but not identical.


Terminology: The abstract as well as the introduction characterizes
the contribution as achieving near-optimal regret against the "optimal
offline allocation policy". This is somewhat imprecise, as the offline
policy can be construed as one that knows the instance (for instance,
in the definition of competitive ratio). However, the guarantees are
against the "offline policy" that knows the distribution but not the
realization. Perhaps a more explicit terminology might be useful.

---

> ### Author Response · Authors · 2022-08-02
> **Thank you for your comments**
>
> Thank you for your comments. Regarding the benchmark we use for regret: the reviewer is correct to point out that the expected case problem can be different from the offline problem with known T instances of item valuations. However, over T rounds, the empirical distribution of the items will be very close to the population distribution with high probability (using Lemma 7 (DKW inequality)). This means that the optimal offline solution with known instances will be very close to the optimal expected case solution.
>
> Regarding the reviewers’ comment on the reliance on iid assumption, we give an intuitive counterexample to the existence of a BIC policy when agents are heterogeneous (independent but not identical). For further discussion on this, please see the response to Question 1 of Reviewer 3.
>
> Under the assumption that the agents are truthful, the online algorithm in fact does satisfy the target distribution constraints exactly (assuming $p^*_i$ and $T$ are chosen such that $p^*_iT$ are integers). Once one of the agents has been allocated $p^*_iT$, the algorithm will simply allocate the rest of the items in an arbitrary order to make sure that every agent receives his target allocation. This will not change the guarantees on the regret bound. We have clarified this part of the algorithm in our revision.

---

> > ### Comment · Reviewer_jFe4 · 2022-08-07
> > **thanks**
> >
> > Thanks for the response; I have read it, and it clarifies my questions.

---

### Meta-Review · Area_Chair_P1Gz · 2022-08-29

**Recommendation:** Accept
**Confidence:** Less certain

**Metareview:**

Executive summary:

The authors study the repeated allocation of an identical good over T rounds to n strategic buyers in a "no monetary transfers" setting. The buyers have i.i.d. valuations drawn from an unknown distribution, and the algorithm must work with reported valuations. The goal is to maximize social welfare (= sum of valuations) under the constraint that each buyer receives a pre-specified fraction of the total number of goods.

The main result is an algorithm for this problem that ensures two things:

(a) approximate Bayesian incentive compatibility (approx-BIC) (Definition 2 and Theorem 1) and

(b) low individual regret (Definition 3 and Theorem 2).

The key idea of the algorithm (is to exploit the iid-ness of the problem) and detect misreports from the underlying CDF using Dvoretzky-Kiefer-Wolfowitz type bounds.

Discussion and recommendation:

After some initial set back on the problem motivation, the reviewers bought into the motivation for studying this online allocation problem "without monetary transfers" (adding examples such as the foodbank example might be good).

There was some discussion around "assuming iid valuations" limiting the generality of the result, but there is in fact a history of papers that studies learning with strategic agents under this assumption (eg Kanoria and Nazerzadeh 2021).

The main difference of the current work is that it's working in a setting without money.

The idea behind the algorithm is maybe "the obvious think to do" - but of course it still requires some work to formally prove that it actually works.

I think one thing that could strengthen the paper would be to add some discussion around the tightness/non-tightness of the approximate BIC and individual regret bounds.

Weak accept.

**Award:**

No

---

### Decision · Program_Chairs · 2022-09-14

Accept